# Memory B cells are reactivated in subcapsular proliferative foci of lymph nodes

Imogen Moran[1,2], Akira Nguyen[1,2], Weng Hua Khoo[3,4], Danyal Butt[1,11], Katherine Bourne[1], Clara Young[1], Jana R. Hermes[1], Maté Biro [5], Gary Gracie[6], Cindy S. Ma[1,2], C. Mee Ling Munier [7], Fabio Luciani[7,8], John Zaunders[7,9], Andrew Parker[6], Anthony D. Kelleher[7,9], Stuart G. Tangye [1,2], Peter I. Croucher[2,3,4], Robert Brink[1,2], Mark N. Read[10] & Tri Giang Phan [1,2]

Vaccine-induced immunity depends on the generation of memory B cells (MBC). However, where and how MBCs are reactivated to make neutralising antibodies remain unknown. Here we show that MBCs are prepositioned in a subcapsular niche in lymph nodes where, upon reactivation by antigen, they rapidly proliferate and differentiate into antibody-secreting plasma cells in the subcapsular proliferative foci (SPF). This novel structure is enriched for signals provided by T follicular helper cells and antigen-presenting subcapsular sinus macrophages. Compared with contemporaneous secondary germinal centres, SPF have distinct single-cell molecular signature, cell migration pattern and plasma cell output. Moreover, SPF are found both in human and mouse lymph nodes, suggesting that they are conserved throughout mammalian evolution. Our data thus reveal that SPF is a seat of immunological memory that may be exploited to rapidly mobilise secondary antibody responses and improve vaccine efficacy.

[1] Immunology Division, Garvan Institute of Medical Research, Sydney, NSW 2010, Australia. [2] St Vincent's Clinical School, Faculty of Medicine, UNSW, Sydney, NSW 2010, Australia. [3] Division of Bone Biology, Garvan Institute of Medical Research, Sydney, NSW 2010, Australia. [4] School of Biotechnology and Biomolecular Sciences, Faculty of Science, UNSW, Sydney, NSW 2052, Australia. [5] EMBL Australia, Single Molecule Science Node, School of Medical Sciences, UNSW, Sydney, NSW 2052, Australia. [6] Department of Anatomical Pathology, St Vincent's Hospital, Sydney, NSW 2010, Australia. [7] The Kirby Institute for Infection and Immunity in Society, UNSW, Sydney, NSW 2052, Australia. [8] School of Medical Sciences, Faculty of Medicine, UNSW, Sydney, NSW 2052, Australia. [9] St Vincent's Hospital Sydney Centre for Applied Medical Research, Sydney, Australia. [10] School of Life and Environmental Sciences and the Charles Perkins Centre, University of Sydney, Sydney, NSW 2052, Australia. [11] Present address: Biologics Research and Development, Teva Pharmaceuticals, Macquarie Park, NSW 2113, Australia. Correspondence and requests for materials should be addressed to T.G.P. (email: t.phan@garvan.org.au)

The concept of immunity dates back to Ancient Greece, with the description by Thucydides in 430BC of the protection afforded to survivors of the Plague of Athens from subsequent reinfection. Since then, vaccines have been empirically developed to harness this power of the immune system to remember past exposures to infectious organisms, and humoral immunity against common viral and vaccine antigens have been shown to provide life-long protection against reinfection[1]. This protection is mediated by neutralising antibodies secreted by long-lived plasma cells (LLPCs) and by memory B cells (MBCs) that proliferate and differentiate more rapidly than naive B cells into antibody-secreting plasma cells upon re-exposure to the antigen[2]. However, despite recent advances in our understanding of MBC heterogeneity, location and functional specialisation[3], the precise question of where they are localised in lymph nodes and how they are reactivated to secrete neutralising antibodies is unknown.

MBCs are strategically positioned outside the B cell follicle at potential sites of antigen drainage, such as the lung following viral infection, the marginal zone in the spleen, the bone marrow and the mucosal epithelium in tonsils (reviewed in ref.[3]). In addition, MBCs accumulate in draining lymph nodes following subcutaneous immunisation[4], where IgG1$^+$ MBCs have been reported to localise adjacent to contracted GCs, whereas IgM$^+$ MBCs are scattered throughout the follicle[5]. The relationship between these tissue resident MBCs and those recirculating in the peripheral blood are still unclear, although a recent study suggests that they are distinct cell types[6].

In the lymph node, the immune response pathways for naive B cell activation in the primary antibody response have been extensively studied. CD169+ subcapsular sinus (SCS) macrophages sample the lymph and present captured antigen on their surface to activate naive B cells[7–10]. Activated B cells migrate to the T-B border[11–13] or interfollicular zone[14] to acquire T cell help, undergo CD40L-dependent proliferation[15] and differentiate into either extrafollicular short-lived plasma cells, or follicular germinal centre (GC) B cells.

Here, we use intravital two-photon microscopy and single-cell RNA sequencing to deconvolute the secondary antibody response and show that the seat of B cell memory lies in a novel structure we have termed the subcapsular proliferative foci (SPF). Reactivated MBCs are shown to proliferate and differentiate into short-lived plasma cells in the SPF, which is anatomically and functionally distinct from the GC. SPF cells differ from GC B cells in terms of their motility, migratory behaviour, single-cell molecular signatures and dependence on BCR signalling for survival. Importantly, we describe similar microanatomical structures in lymph nodes from patients, demonstrating that this is an evolutionarily conserved immune response pathway.

## Results

**Resting MBCs reside in a subcapsular niche**. To determine the immune response pathways involved in MBC reactivation, we adoptively transferred SW$_{HEL}$ B cells[16] expressing the optical highlighter Kaede[17] and OT2 T cells[18], and immunised recipient mice with the cognate antigen hen egg lysozyme (HEL) conjugated to ovalbumin (OVA). Mice were analysed >28 days later when the primary antibody response has resolved and antigen-specific cells are no longer proliferating (Supplementary Figure 1). After this time point, there are no persisting GCs, as demonstrated by fluorescence-activated cell sorting (FACS) analysis (Supplementary Figure 1). MBCs are able to survive independent of antigen-derived BCR signals[19] and T cell help[20,21], unlike GC B cells which are dependent on both[22,23]. T cell depletion experiments and inducible deletion of

MHCII in responding B cells had no impact on the survival of these cells >28 days after primary immunisation (Supplementary Figure 1). Furthermore, inhibition of BCR signalling with the small molecule ibrutinib also did not impact on their survival (Supplementary Figure 1). Finally, these cells consisted of IgM$^+$ and IgG$^+$ cells that expressed Fas, CD80, CD86, PD-L2, CCR6, CD69, CD62L, EBI2 and CXCR3, but not S1PR1 (Supplementary Figure 2), consistent with the phenotype of murine MBCs[24]. Taken together, these data confirm that antigen-specific cells persisting >28 days after primary immunisation were bona fide MBCs and not GC B cells.

Analysis of whole-mount lymph nodes by two-photon microscopy (TPM) 70 days later revealed that while B cells were detectable in the follicle as deep as 300 μm below the surface of the capsule (Supplementary Figure 3), antigen-specific SW$_{HEL}$ donor-derived B cells were preferentially positioned in the outer B cell follicle, where they were closely associated with CD169$^+$ SCS macrophages (Fig. 1a and Supplementary Movie 1). This localisation of MBCs near sites of antigen drainage is similar to the positioning of follicular memory T cells that also patrol the subcapsular region[25].

To analyse the migration and behaviour of these MBCs by intravital imaging, we set up recipient mice in which SW$_{HEL}$ MBCs expressed tdTomato and adoptively transferred naive B cells the day before imaging. This showed that MBCs, compared to naive B cells, were largely confined to the subcapsular region (Fig. 1b and Supplementary Movie 2). Analysis of the cell tracks and positions relative to CD169-labelled SCS macrophages showed that at any particular point in time ~80% of MBCs were located in proximity (<20 μm) to SCS macrophages (Fig. 1c). In contrast, <50% of naive B cells in the imaging volume were located in proximity to SCS macrophages. Time-lapse images showed that MBCs made frequent serial contacts with SCS macrophages, consistent with antigen surveillance behaviour (Fig. 1d and Supplementary Movie 3). In addition, MBCs in close proximity (<20 μm) to SCS macrophages had a decreased average speed compared to MBCs further away (>20 μm) from SCS macrophages (Fig. 1e) and MBCs had an increased arrest coefficient compared to naive B cells (Fig. 1f). These findings further support the conclusion that MBCs in the subcapsular region are scanning SCS macrophages in search of antigen.

We next enumerated the number of MBCs and found that there were twice as many MBCs in draining compared to non-draining lymph nodes (Fig. 1g). To determine the residence time, we photoconverted MBCs expressing the optical highlighter Kaede and tracked the ratio of resident photoconverted to newly arrived non-photoconverted cells in the lymph node (Fig. 1h). This cell tracking showed that MBCs resided in the lymph node with a half-life of 48 h (Fig. 1i), which equates to a lymph node residence time of 69 h, compared to <24 h for naive B cells[26]. These data show that MBCs preferentially accumulate in the draining lymph node where they reside in a subcapsular niche and interact with SCS macrophages.

**MBCs and naive B cells present similar migratory behaviours**. Both naive and MBCs must search the lymph node for scarce antigen to be able to respond efficiently. We therefore next analysed the motility and migratory patterns of naive and MBCs to determine if they employed similar search strategies. Interestingly, cell tracking and image analysis showed that both cell populations migrated with a similar median speed (Fig. 2a), displacement (Fig. 2b) and meandering index (Fig. 2c). It should be noted that we detected very small differences in these parameters (as measured by the Kolmogorov–Smirnov $D$ statistic) which were statistically significant because of the large number of

cells that were tracked; however, these differences are unlikely to be of biological significance (for details see Methods). Analysis of the mean squared displacement (MSD) plot of the cell trajectories (Fig. 2d) showed that both naive and MBCs migrated with an $\alpha$ slope >1, consistent with a superdiffusive migration pattern. Model fitting of the cell trajectories using multi-objective optimisation[27] showed that the motility of MBCs and naive B cells was best captured as heterogeneous correlated random walks (CRWs) (Supplementary Figure 4 and Methods). Accordingly, both had very similar representative track profiles (Fig. 2e). Thus, naive and MBCs patrol the subcapsular region of the lymph node using similar search strategies.

**MBCs cluster in the subcapsular region.** To determine if the subcapsular region was also the site of MBC reactivation, we next imaged the in situ immune responses to antigen recall. In this instance, we adoptively transferred tdTomato SW$_{HEL}$ B cells and cyan OT2 T cells and immunised with HEL-OVA to establish fluorescent MBCs that can be tracked. Mice were re-immunised >28 days later in the memory phase. On day 5 post-recall, intravital two-photon microscopy revealed that reactivated SW$_{HEL}$ MBCs segregated into two distinct clusters, one in the subcapsular region and the other in secondary GCs (Fig. 3a and Supplementary Movie 4). Analysis of the migration pattern of reactivated MBCs at this time point showed that these two cell

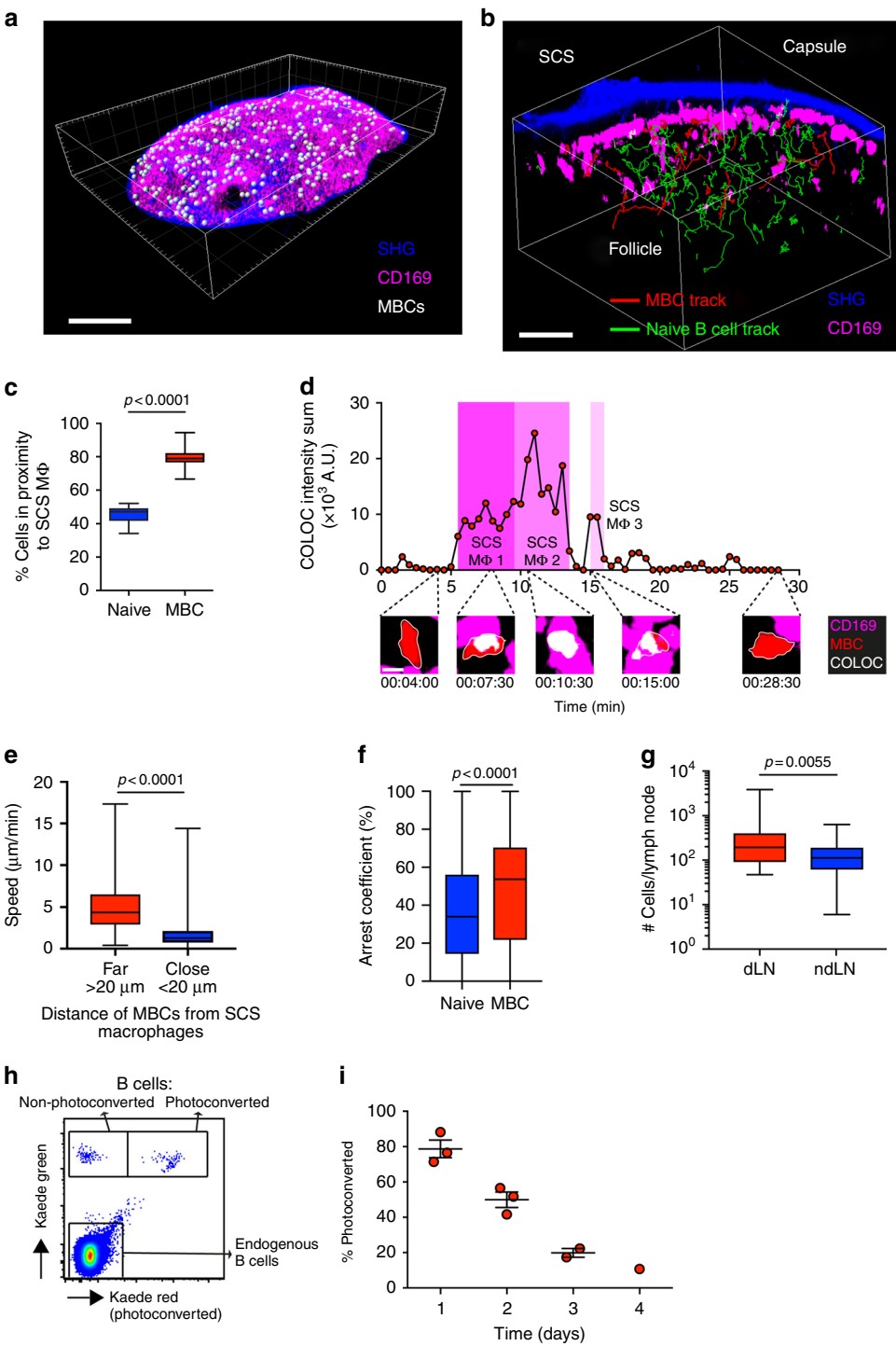

populations were spatially confined to their respective micro-anatomical compartments, and only rarely migrated into the follicular mantle zone (Fig. 3b).

**MBCs interact with Tfh cells in the subcapsular region**. Interestingly, the subcapsular region and follicular mantle zone also contained a large number of secondary Tfh cells that migrated in a pattern that was also best captured as a superdiffusive CRW (Supplementary Figure 5). We have previously described the presence of secondary Tfh cells in the subcapsular region where they interacted with SCS macrophages[25]. However, in those studies we did not image the B cell compartment. In the current study, we show that these secondary Tfh cells also interacted with B cells in this region, as evidenced by the frequent short-lived contacts between cyan OT2 Tfh cells and $SW_{HEL}$ MBCs (Fig. 3c and Supplementary Movie 5). These interactions are reminiscent of the brief T-B entanglements described inside GCs during primary antibody responses[28,29], and suggest that these are productive T-B collaborations. To further characterise these T-B interactions, we also imaged the primary antibody response on day 5, when both T and B cells have colonised the follicle[25]. These data showed that the T-B interactions were similar in number (Fig. 3d) and duration (Fig. 3e) between the primary and secondary responses. These data confirm that the subcapsular region is a previously unappreciated site of T cell–B cell interaction.

**MBCs proliferate and differentiate into plasma cells in SPF**. To further characterise the B cell cluster in the subcapsular region, we harvested lymph nodes on days 3 to 5 post-recall for histological analysis. This revealed that cells in the subcapsular cluster were cycling as demonstrated by staining for the proliferation marker Ki67, a property they shared with cells in the GC (Fig. 4a, b). We have therefore named this novel structure the subcapsular proliferative foci (SPF). However, in contrast to the GC, there was no staining for $CD35^+$ follicular dendritic cells in the SPF. We also noted that a large fraction of SPF cells demonstrated more intense cytoplasmic staining for antibodies to HEL than GC B cells (Fig. 4c), suggesting that they were antibody-secreting plasma cells. To confirm this, we set up secondary antibody responses with $SW_{HEL}$ B cells carrying the Blimp-1$^{egfp/+}$ reporter to identify plasma cells in vivo[30]. As early as day 3 post-recall, analysis revealed large clusters of GFP expressing cells in the subcapsular region (Fig. 4d). We also performed histological analysis for Bcl-6 and showed that HEL-binding cells in the SPF did not express this transcription factor, further highlighting their

differences from GC B cells (Fig. 4e, f). Thus, MBCs proliferate and differentiate into plasma cells in the SPF.

**Two distinct subpopulations of SPF cells**. Close examination of the migratory behaviour of B cells in the SPF (Supplementary Movie 4) suggested that they were heterogeneous in terms of their motility. Therefore we set up tdTomato $SW_{HEL}$ MBCs carrying the Blimp-1$^{egfp/+}$ reporter and re-immunised mice to determine if there were differences in cell migration and if this tracked with Blimp-1 expression. Indeed, we observed two patterns of B cell migration in the SPF based on Blimp-1 expression (Fig. 5a, b, and Supplementary Movie 6). Blimp-1$^+$ cells were shown to be rounded, largely immotile and arrested on the surface of SCS macrophages, while Blimp-1$^{neg}$ cells were highly motile and migrated with much greater speeds (Fig. 5c), displacement (Fig. 5d) and meandering indices (Fig. 5e). Notably, the differences in these parameters were both biologically (Kolmogorov–Smirnov D statistic) and statistically (Kolmogorov–Smirnov p value) significant. Analysis of the MSD plot of the cell trajectories showed that Blimp-1$^+$ cells migrated with an $\alpha$ slope <1, consistent with an anomalous subdiffusive migration pattern (Fig. 5f). This search strategy has been suggested to increase the probability of finding nearby targets, particularly in areas where the target is abundant[31], as might occur for antigen trapped by SCS macrophages. In contrast, Blimp-1$^{neg}$ cells migrated with an $\alpha$ slope >1, consistent with a superdiffusive migration pattern, similar to that for naive and MBCs in the steady state (Fig. 2d). This difference in motility is apparent in the $xy$ displacement patterns of representative tracks from both subpopulations (Fig. 5g). Nevertheless, model fitting of the cell trajectories using multi-objective optimisation showed that the motility of Blimp-1$^+$ and Blimp-1$^{neg}$ cells could both be modelled as inverse heterogeneous CRWs (Supplementary Figure 6 and Methods). These data show that SPF cells could be separated into two distinct populations based on their motility and Blimp-1 expression.

**Single-cell RNA sequencing of the memory response**. To further delineate the composition of the secondary antibody response, we performed single-cell RNA sequencing of 142 FACS-sorted donor-derived $SW_{HEL}$ B cells on day 5 post-recall (Supplementary Figure 7). Of these, 122 cells passed sequence and cell QC. Bayesian analysis of single-cell gene expression data revealed 8190 differentially expressed genes (Fig. 6a), and non-negative matrix factorisation (NMF) established that responding B cells could be

**Fig. 1** MBCs scan SCS macrophages for antigen. **a** Tiled maximum intensity projection (3072 × 2048 × 141 μm) of lymph node 70 days after HEL-OVA primary immunisation. MBC spots (white), $CD169^+$ SCS macrophages (pink), SHG (blue). See also Supplementary Movie 1. Scale bar = 500 μm. **b** Maximum intensity projection (300 × 300 × 48 μm) of follicle 32 days after HEL-OVA immunisation. MBC $SW_{HEL}$ tracks (red), naive $SW_{HEL}$ B cell tracks (green), $CD169^+$ SCS macrophages (pink), SHG (blue). See also Supplementary Movie 2. Scale bar = 50 μm. **c** Proportion of naive B cells (blue) or MBCs (red) in proximity (<20 μm) to $CD169^+$ SCS macrophage surface at any given time point, number time points = 46, quantified from **b** where number naive cells = 74, number MBCs = 40. Box plots show centre line as median, box limits as upper and lower quartiles, whiskers as minimum to maximum values. **d** Maximum intensity projection of follicle 54 days after HEL-OVA primary immunisation, showing colocalisation (white) of a MBC (red) with $CD169^+$ SCS macrophages (pink) over time. Interactions with three different SCS macrophages are highlighted in shades of pink. See also Supplementary Movie 3. Scale bar = 5 μm. **e** Speed of MBCs segmented based on distance from $CD169^+$ SCS macrophage surface: far (<20 μm) (red) or close (>20 μm) (blue). Number of data points = 1827. Data are combined from 4 movies. Comparison between groups utilised one-tailed unpaired Student's t-test. **f** Arrest coefficient of naive B cells (blue) or MBCs (red), defined as percentage of time a cell slowed down to <3μm/min. Number of naive cells = 733, MBCs = 202. Data are pooled from 4 movies. Comparison between groups utilised one-tailed unpaired Student's t-test. **g** Number of donor-derived MBCs in draining inguinal lymph node (dLN) (red) compared to non-draining popliteal lymph node (ndLN) (blue). Data are from 15 mice and representative of 2 independent experiments. Comparison between groups utilised one-tailed unpaired Student's t-test. **h** Identification of unphotoconverted Kaede green MBCs, photoconverted Kaede red MBCs and endogenous B cells in the draining lymph node by FACs analysis, example shown is 2 days after photoconversion. Box plots show centre line as median, box limits as upper and lower quartiles, whiskers as minimum to maximum values. **i** Kaede MBCs in the draining lymph node were photoconverted, allowed to recover from anesthesia and then harvested 1–4 days later. Graph shows proportion of photoconverted cells to non-photoconverted cells in the lymph node at indicated time points

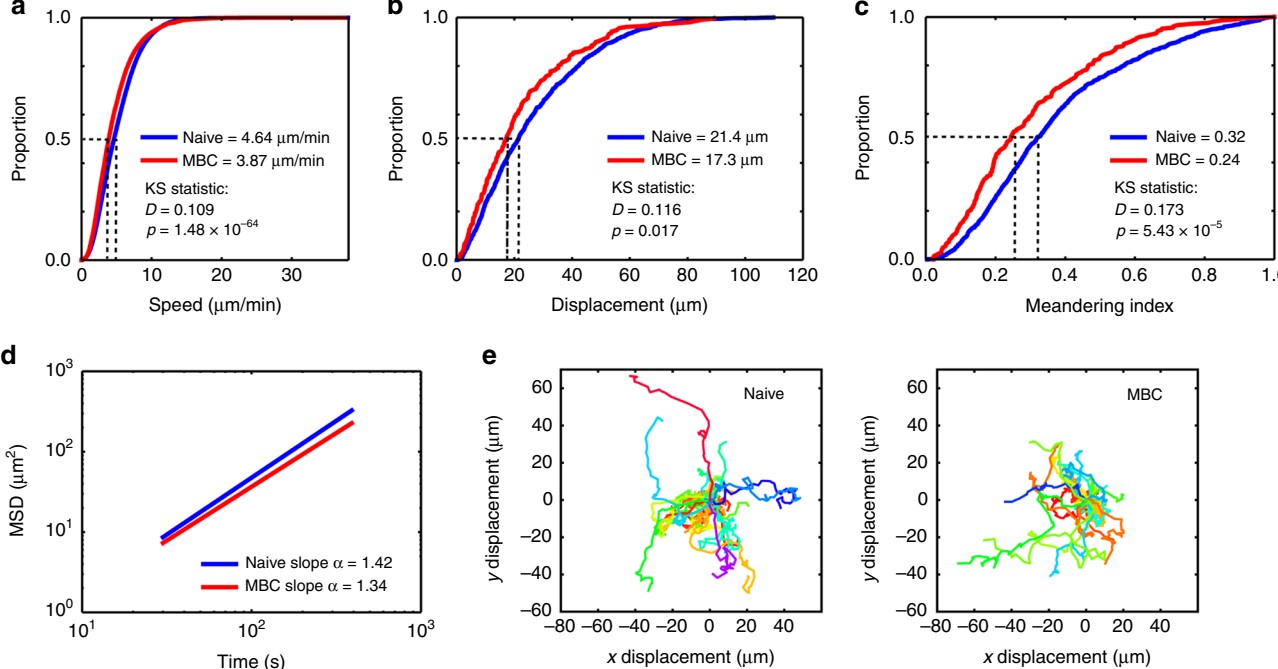

**Fig. 2** Naive and MBCs use a superdiffusive search strategy for antigen. **a** Cumulative distribution function (CDF) plot of speed for naive (blue) and MBCs (red). **b** CDF plot of displacement for naive (blue) and MBCs (red). **c** CDF plot of meandering index for naive (blue) and MBCs (red). KS statistics refer to Kolmogorov–Smirnov statistics. **d** Mean squared displacement (MSD) plot of naive (blue) and MBCs (red). All data are pooled from three independent movies. **e** Representative tracks for naive (left) and MBCs (right). Data are representative of >3 independent experiments

most robustly and reproducibly deconvoluted into three discrete subpopulations (cophenetic coefficient = 0.999) (Fig. 6b). Population 1 was characterised by expression of genes related to somatic hypermutation, cell division and mitosis that are typically associated with GC B cells such as *Aicda*, *Mki67* and *Ccnb1* (Fig. 6c, Supplementary Figure 8 and Supplementary Data 1). Population 2 was characterised by expression of genes typically associated with endoplasmic reticulum stress and antibody secretion by plasma cells[32] such as *J chain*, *Spcs3* and *Xbp1* (Fig. 6c, Supplementary Figure 8 and Supplementary Data 2). Population 3 was characterised by expression of genes typically associated with MBC trafficking and cell positioning in the outer follicle such as *Sell*, *Ccr6* and *Gpr183* (Fig. 6c, Supplementary Figure 8 and Supplementary Data 3). We next performed gene set enrichment analysis (GSEA) to confirm the identity of these three populations. This showed that population 1 was highly enriched for genes associated with GC (Fig. 6d and ref.[33]), population 2 for genes associated with plasma cell (Fig. 6e and ref.[34]) and population 3 for MBC signature genes (Fig. 6f and ref.[34]). Thus, in agreement with the intravital imaging, the SPF can be readily resolved into plasma cells and a second population of SPF B cells with features of MBCs by single-cell RNA sequencing.

Using differentially expressed genes from the single-cell transcriptomic analysis, we were able to resolve these same three cell populations by both surface and intracellular FACS analysis, thereby enabling precise tracking of the SPF. Hence, population 1 were CD38$^{lo}$ cells that expressed low levels of immunoglobulin and were Bcl-6 positive; population 2 were CD38$^{int}$ cells that expressed intermediate levels of immunoglobulin and were Bcl-6 negative; and population 3 were CD38$^{hi}$ cells that expressed high levels of intracellular immunoglobulin and were Bcl-6 negative (Fig. 6g). Alternatively, population 1 could be identified by surface staining as CD38$^{lo}$CD138$^{neg}$B220$^{hi}$CCR6$^{neg}$ GC B cells; population 2 as CD38$^{lo}$CD138$^{+}$B220$^{lo}$CCR6$^{neg}$ plasma cells; and population 3 as CD38$^{hi}$CD138$^{neg}$B220$^{hi}$CCR6$^{+}$ MBCs (Supplementary Figure 9). Notably, both population 2 and 3 expressed

CXCR3 and Sca-1 (Supplementary Figure 9) indicating that these surface markers can also be used to discriminate between SPF and GC B cells.

**Pseudotime analysis of lineage trajectories**. To further characterise the relationship between these three cell populations, we performed pseudotime analysis to infer their lineage trajectories[35]. This showed that a large distance separated population 1 from population 2 and 3, which were closely spaced in pseudotime (Fig. 6h). Thus, in accordance with the imaging data, responding B cells can be separated into GC and SPF cells, and the latter separated again into plasma cell and SPF cells with MBC-like properties. As plasma cells are terminally differentiated, this pseudotemporal ordering also suggests the population 2 plasma cells can arise from a subset of population 3 SPF B cells. To further determine the origin of population 2 plasma cells, we compared the secondary antibody response on day 5 and 7. This showed that, consistent with their expression of CXCR3, the plasma cells were short-lived and decreased between day 5 and 7 (Fig. 6i). At the same time there was an increase in the proportion of population 1 GC B cells and no change in the proportion of population 3 SPF B cells. This suggests that the plasma cells in the SPF are unlikely to be derived from secondary GCs, as one would expect a concomitant increase in plasma cells as the GC expands and matures[36]. Thus, the imaging and single-cell RNA sequencing confirm that short-lived plasma cells are generated from MBCs in the SPF.

**SPF cell survival independent of Btk-mediated BCR signalling**. To test the role of T cell help in the generation of the SPF, we set up recipient mice with SW$_{HEL}$ MBCs carrying the *H2-Ab1*$^{fl/fl}$. *Cd79a*.Cre$^{ERT2}$ alleles to enable inducible deletion of MHCII in MBCs by tamoxifen administration >28 days after primary immunisation. Mice were then re-immunised 7 days after tamoxifen treatment (Fig. 7a). Compared to the mice administered

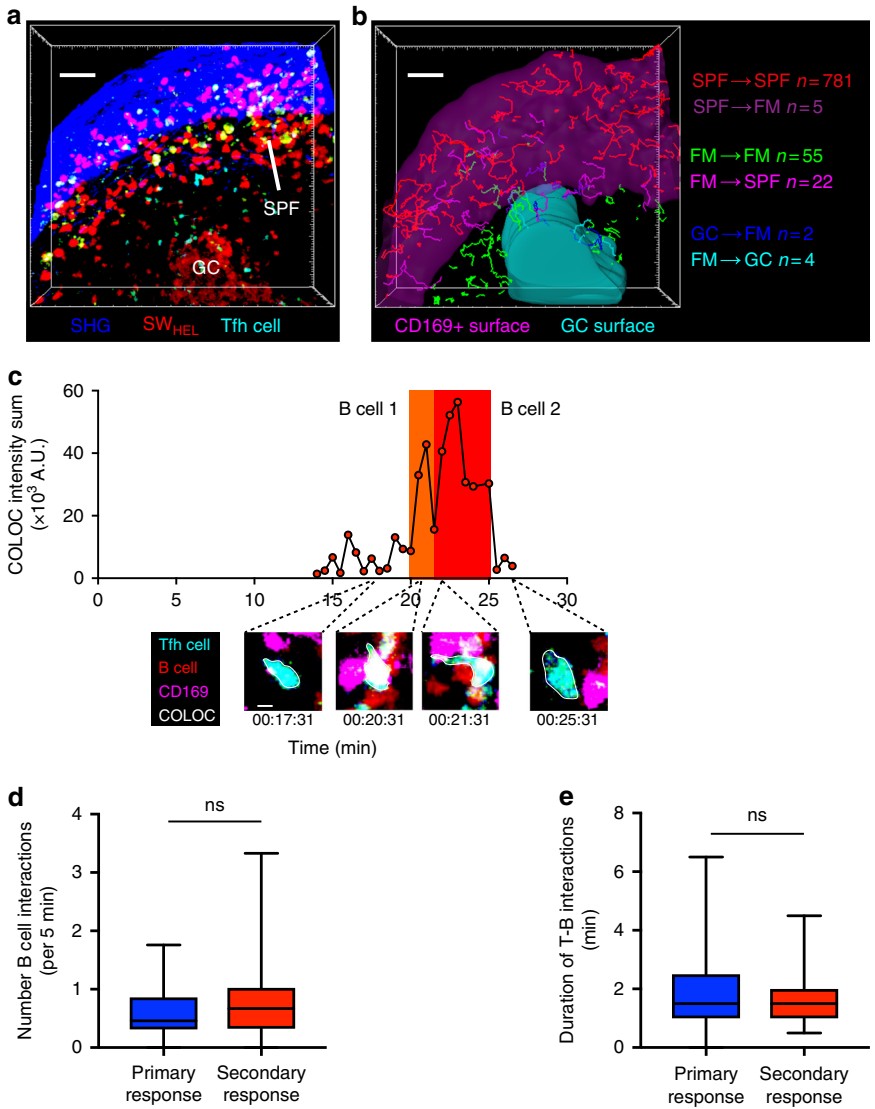

**Fig. 3** MBCs cluster and interact with Tfh cells in subcapsular foci. **a** Maximum intensity projection (504 × 465 × 31 μm) of follicle imaged 5 days after HEL-OVA secondary immunisation. SW$_{HEL}$ B cells (red), OT2 T cells (cyan), SHG (blue). Scale bar = 50 μm. **b** Cell tracking of SW$_{HEL}$ B cell from **a** with tracks colour coded based on migration pattern. CD169$^+$ subcapsular region surface (pink) and GC surface (cyan). Representative 100 tracks out of a total of 781 in the subcapsular region are shown. See also Supplementary Movie 4. Scale bar = 50 μm. **c** Time-lapse images 5 days after secondary immunisation, showing colocalisation (white) of Tfh cells (cyan) with SW$_{HEL}$ B cells (red). CD169$^+$ SCS macrophages (pink), SHG (blue). Scale bar = 5 μm. See also Supplementary Movie 5. **d** Number of times a T cell interacts with a B cell per 5 min, 5 days after primary immunisation (blue) or secondary immunisation (red). Data are combined from 2 movies in the primary response, $n = 30$, and 4 movies in the secondary response, $n = 33$. Comparison between groups utilised one-tailed unpaired Student's $t$-test. **e** Duration of T-B interactions 5 days after primary immunisation (blue) or secondary immunisation (red). Data is combined from 2 movies in the primary response, $n = 49$, and 4 movies in the secondary response, $n = 49$. Comparison between groups utilised one-tailed unpaired Student's $t$-test. Box plots show centre line as median, box limits as upper and lower quartiles, whiskers as minimum to maximum values

vehicle control, this lead to significant reductions in all three cell populations, indicating that SPF B cells and plasma cells, similar to GC B cells, required T cell help for their development.

To delineate the role of BCR signalling in the maintenance of the SPF response, we re-immunised mice and 2 days later, when the SPF is established, administered the Btk small molecule inhibitor ibrutinib (Fig. 7b). This resulted in a significant reduction in the number of GC B cells, but had no effect on the number of SPF B cells or plasma cells, suggesting that cells in the SPF were not dependent on Btk-mediated BCR signals for their survival. Since ibrutinib treatment had no effect on survival of MBCs (Supplementary Figure 1), this suggests that Btk-mediated BCR signalling is required for initial activation of MBCs but not for the survival of MBCs in the SPF, whereas it is required

for survival of GC B cells. This is consistent with clinical data showing that ibrutinib depletes pre-germinal centre B cells but not isotype-switched memory B cells in patients with chronic lymphocytic leukaemia[37]. Since ibrutinib may affect the initial activation of MBCs, we also administered ibrutinib before and during activation of the secondary response (Supplementary Figure 10). As expected, this resulted in a significant reduction in the number of GC B cells as well as SPF B cells and a statistically non-significant reduction in the number of plasma cells. Thus, Btk-mediated BCR signalling is required for MBC reactivation and the survival of GC B cells, but not the survival of SPF B cells and plasma cells.

Finally, we asked if this novel structure was unique to the mouse or if it is also present in human lymph nodes. Analysis of

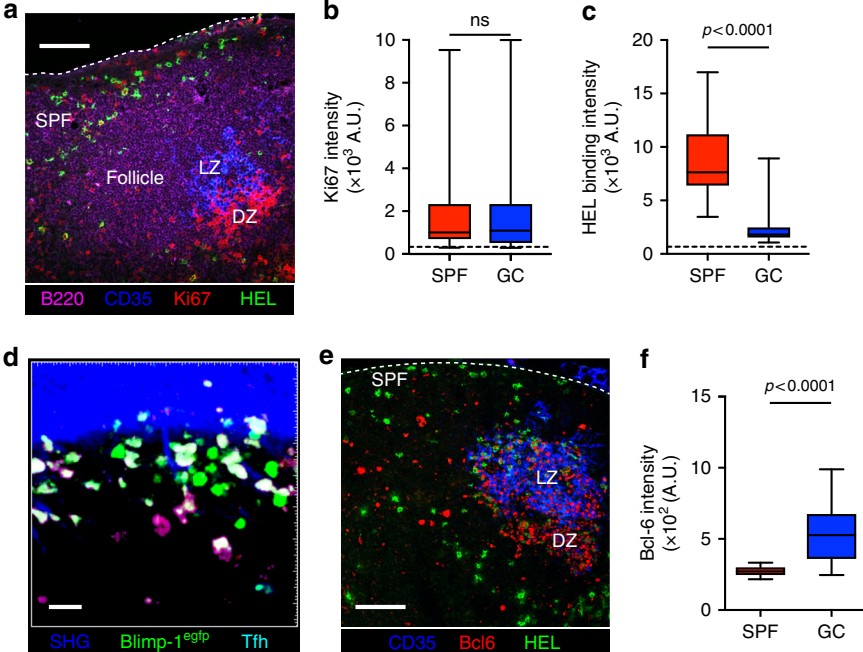

**Fig. 4** MBCs differentiate into plasma cells in SPF. **a** Representative histological section of lymph node 3 days after secondary immunisation. CD35 (blue), Ki67 (red), HEL binding (green), B220 (pink). Scale bar = 100 μm. **b** Quantification of Ki67 intensity on HEL-binding cells in SPF, n = 57, or GC, n = 72, from **a**. Comparison between groups utilised Mann–Whitney U test. **c** Quantification of HEL-binding intensity on HEL-binding cells in SPF, n = 57, or GC, n = 72, from **a**. Comparison between groups utilised Mann–Whitney U test. **d** Orthogonal maximum intensity projection (200 × 200 × 10 μm) 3 days after secondary immunisation showing Blimp-1$^+$ cells (green), Tfh cells (cyan), SHG (blue). Note autofluorescent macrophages. Scale bar = 20 μm. **e** Representative histological section of lymph node 5 days after HEL-OVA re-immunisation. CD35 (blue), Bcl6 (red), HEL binding (green). Scale bar = 100 μm. **f** Quantification of Bcl-6 intensity on HEL$^+$ cells in SPF B cells, n = 26, or GC B cells, n = 46. Data are representative of 5 independent experiments. Comparison between groups utilised Mann–Whitney U test. Box plots show centre line as median, box limits as upper and lower quartiles, whiskers as minimum to maximum values

serial sections from inflamed lymph nodes revealed clusters of CD138$^+$ plasma cells in the subcapsular region in contact with CD169$^+$ SCS macrophages in 3/3 patients (Fig. 7c). This distribution of plasma cells in the lymph node follicle of patients with ongoing immune responses was strikingly similar to the images of the SPF in murine lymph nodes (Fig. 4a).

## Discussion

Long-lived humoral immunity is initially dependent on the secretion of neutralising antibodies by LLPCs. In the event of inadequate antibody titres, MBCs provide a critical second line of defence by rapidly proliferating and differentiating into plasma cells. It was previously noted that MBCs often localised to extrafollicular sites that are exposed to antigen, such as the splenic marginal zone, and that this efficient access to antigen was proposed to contribute to the rapid kinetics of the secondary antibody response[38]. However, it is unclear what supporting cells and structures are available in these sites to capture and present antigen and also provide T cell help. Here, we have focussed on the localisation and reactivation of MBCs in skin draining lymph nodes following subcutaneous immunisation. Our data show that resting MBCs preferentially accumulate in the draining lymph node where they reside in close proximity to CD169$^+$ SCS macrophages that are specialised to capture and present antigen to B cells[39–41]. This same niche is also occupied by follicular memory T cells[25], innate-like lymphocytes[42] and NKT cells[43] that are similarly well-adapted to provide the co-stimulatory signals, such as CD40L, and cytokines, such as IL-4[44], that are needed to expedite the secondary antibody response. Notably, we show that, upon antigen recall, MBCs cluster and organise into a novel

structure, the SPF, where antigen and T cell help is concentrated to facilitate plasma cell differentiation.

To study the dynamics of MBC localisation and reactivation in situ, we have established an experimental system in which, after 28 days, the primary antibody response has resolved and there are no persisting GCs. This was confirmed by the absence of Ki67 staining, the CD38$^{hi}$ phenotype of the MBCs and their survival independent of Btk-derived BCR signals and T cell help. The survival of MBCs in our system following ibrutinib treatment is in agreement with their antigen-independence[19], but contrasts with recent data showing MBC survival is dependent on Syk. However, Btk only mediates active antigen-derived BCR signals, whereas Syk mediates both active and 'tonic' BCR signals independent of antigen, possibly downstream of its activation by Fc receptors, integrins or C-type lectins[45], and this may account for the discrepancy. Consistent with this, Btk has recently been shown to be dispensable for mature B cell survival[46] and ibrutinib treatment has been shown to deplete pre-GC B cells but not MBCs in patients with chronic lymphocytic leukaemia[37].

Intravital imaging of resting MBC dynamics show that they adopt a CRW and superdiffusive migration pattern, similar to that for naive B cells. In a CRW, cells migrate with a directional persistence or bias that may reflect the influence of chemokine gradients. In this regard, it is notable that MBCs in the lymph node express the chemokine receptors CXCR3, which is associated with the positioning of cells in the periphery of the lymph node[47], EBI2, which is associated with positioning of cells in the outer follicle[48,49], and CCR6, which is associated with the positioning of cells near the SCS[50]. By using a superdiffusive search strategy, MBCs are able to migrate over large distances and cover large areas in short periods of time, and this optimises the

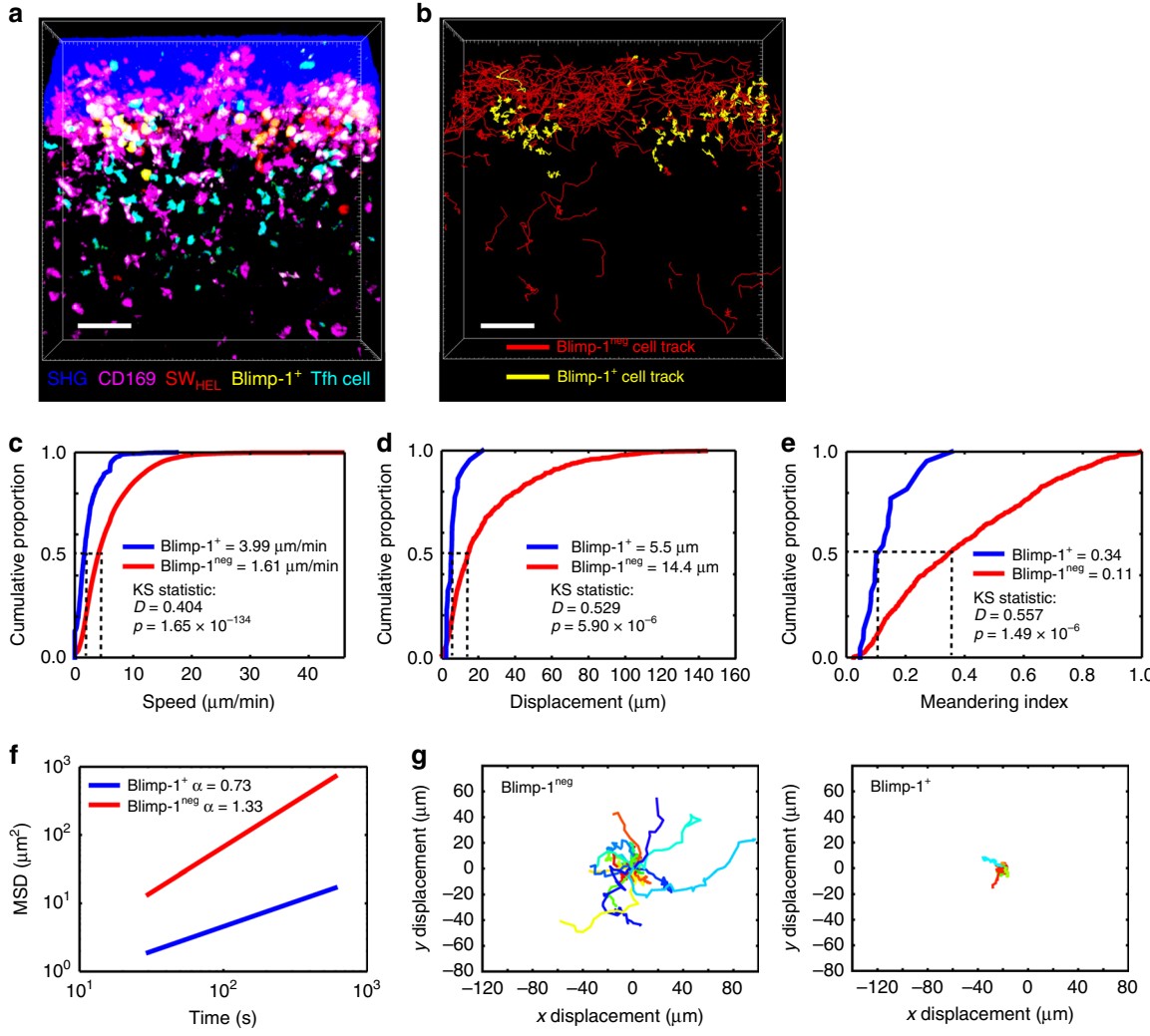

**Fig. 5** Blimp-1 expression identifies two populations of SPF cells. **a** Maximum intensity projection (360 × 360 × 20 μm) 5 days after HEL-OVA secondary immunisation. $SW_{HEL}$ B cells (red), Blimp-1+ $SW_{HEL}$ B cells (yellow, from co-expression of red and green), Tfh cells (cyan), CD169+ SCS macrophages (pink), SHG (blue). Scale bar = 50 μm. **b** Cell tracking analysis of **a**. Blimp-1+ $SW_{HEL}$ B cell tracks (yellow), Blimp -1neg $SW_{HEL}$ B cell tracks (red) in the SPF. See also Supplementary Movie 6. Scale bar = 50 μm. **c** CDF plot of speed for Blimp-1+ (blue) and Blimp -1neg $SW_{HEL}$ B cells (red). **d** CDF plot of displacement Blimp-1+ (blue) and Blimp -1neg $SW_{HEL}$ B cells (red). **e** CDF plot of meandering index for Blimp-1+ (blue) and Blimp -1neg $SW_{HEL}$ B cells (red). KS statistics refer to Kolmogorov–Smirnov statistics. **f** Mean squared displacement (MSD) plot of Blimp-1+ (blue) and Blimp -1neg $SW_{HEL}$ B cells (red). **g** Representative tracks for Blimp-1+ and Blimp -1neg cells in the SPF. Data are representative of 5 independent experiments

efficiency of their search for rare cognate antigens[51,52]. Our data also indicate that MBCs reside in the lymph node with a residence time of 69 h; this lymph node retention is probably mediated, in part, by their CD69 expression, S1PR1 downregulation and desensitisation to S1P gradients[53].

We show that upon re-exposure to antigen, MBCs clustered and organised themselves around two transient structures, the GC and the SPF. The SPF is a large surface that extends along the floor of the SCS and consists of B cells, Tfh cells and CD169+ SCS macrophages. Intravital imaging showed that B cells were spatially confined to these two microanatomical compartments and did not cross over from one to the other. Specifically, we did not observe migration of cells from the GC into the SPF, suggesting that SPF cells do not originate from the GC. Within the SPF, secondary Tfh cells were observed to interact with B cells, indicating that this is an important site for productive T cell–B cell interaction. We have previously shown that follicular memory T cells are activated by SCS macrophages and undergo rapid intrafollicular expansion in the subcapsular region to generate secondary Tfh cells[25]. Our new data suggest that these secondary

Tfh cells localise in the SPF to provide T cell help to SPF B cells. While proliferation of B cells at the perimeter of the follicles in the spleen had been previously described in the primary antibody response[54], under these circumstances naive B cells are still required to migrate to the T-B border to acquire T cell help. Thus, this efficient engineering of the secondary response, whereby antigen and T cell help is concentrated in one site, contrasts with the primary response, where naive B cells must first be activated by antigen in the subcapsular region[8–10] before they are permitted to migrate to the T cell–B cell border[15] and interfollicular regions[14] to acquire T cell help.

While both the GC and SPF were sites of proliferation, there are several key differences between the SPF and the GC in terms of location, structural organisation and function. Firstly, the GC is located deeper in the B cell follicle, whereas the SPF is peripherally located just beneath the SCS. Secondly, unlike the GC, the SPF is Bcl-6 negative and is not organised into light zones containing FDCs and dark zones containing proliferating cells. Thirdly, and perhaps most importantly, the SPF is a site of high plasma cell output where up to 60% of the cells are plasma cells.

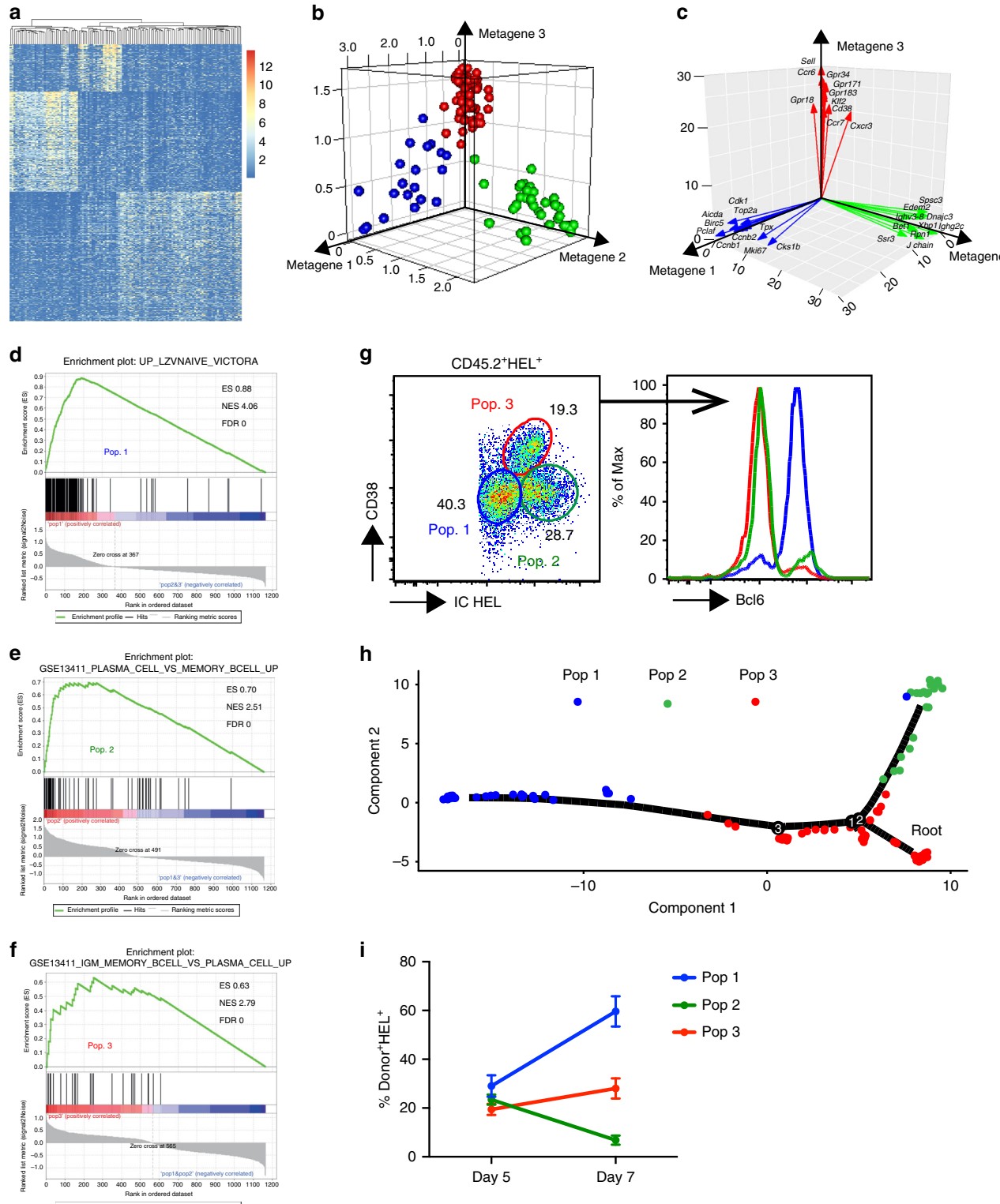

**Fig. 6** Deconvolution of the secondary antibody response by single-cell RNA sequencing. **a** Heatmap showing 8190 differentially expressed genes from 122 single SW$_{HEL}$ donor-derived B cells. **b** Deconvolution of 122 single cells into three populations by expression of metagenes 1, 2 and 3 by NMF. **c** Vector diagram showing individual gene contributions to the metagenes 1, 2 and 3. **d** GSEA identifies population 1 as GC. **e** GSEA identifies population 2 as plasma cell. **f** GSEA identifies population 3 as MBCs. **g** Flow cytometric analysis of donor-derived HEL-binding B cells 4 days after HEL-OVA re-immunisation showing CD38, intracellular (IC) HEL-binding immunoglobulin and Bcl-6 expression on the 3 populations. **h** Pseudotime analysis of single-cell RNA sequencing data. **i** Proportion of 3 populations by flow cytometric analysis of donor-derived HEL-binding B cells 5 and 7 days after HEL-OVA re-immunisation. Data are representative of two independent experiments with 5 mice per time point

In contrast, plasma cell output from the GC is very low due to stringent quality controls that limit plasma cell differentiation to <3% of the high affinity GC B cells[55]. These data suggest that the function of the SPF is to rapidly generate large numbers of antibody-secreting plasma cells in order to swiftly neutralise the offending pathogen. On the other hand, the function of secondary GCs is to further mature the BCR and generate high affinity

plasma cells, a process that may be too slow to prevent overwhelming infection. Interestingly, SPF plasma cells express CXCR3, suggesting that they are short-lived plasma cells[32]. This was confirmed by the decline in plasma cells from day 5 to 7 of the secondary antibody response.

Intravital imaging of cellular dynamics in the SPF revealed two distinct cell migration patterns which correlated with Blimp-1

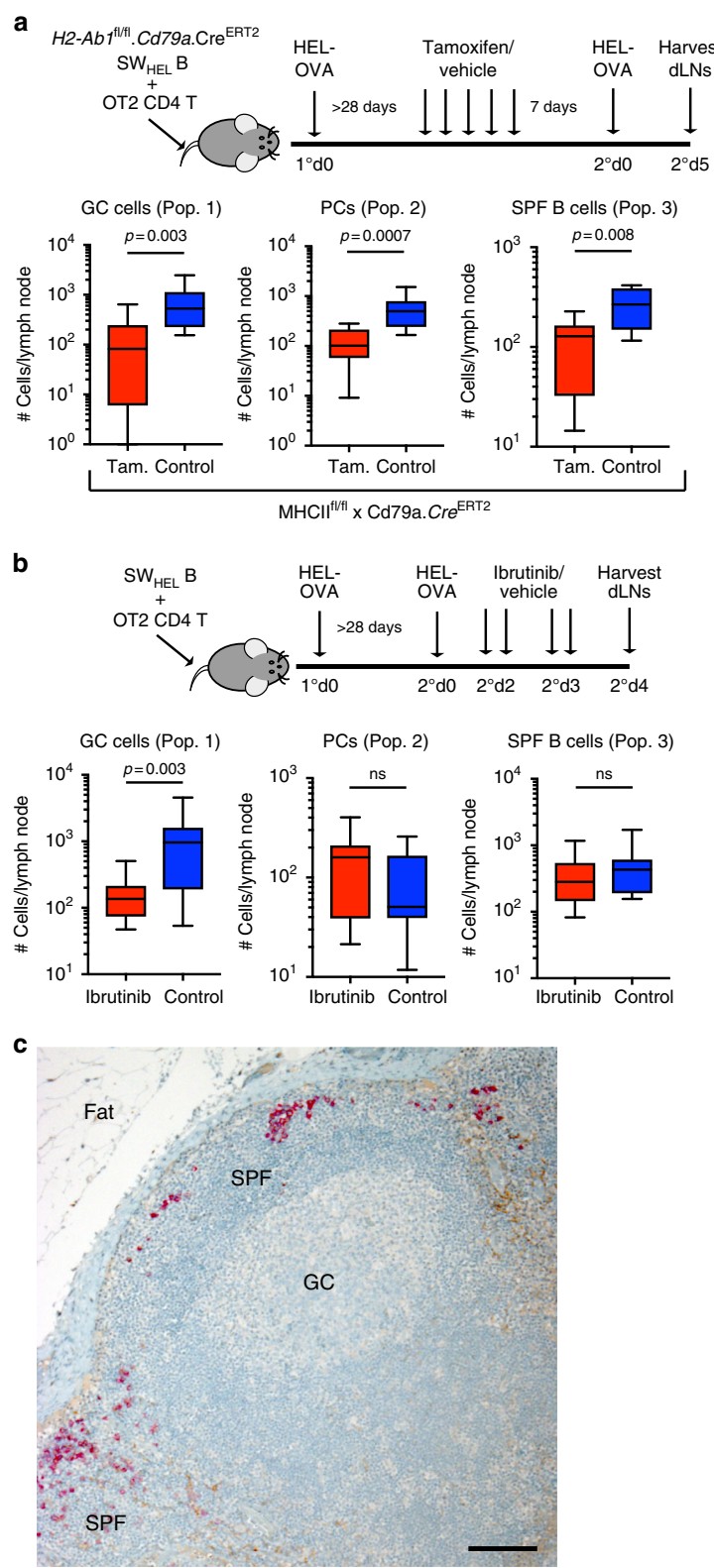

Red = CD138    Brown = CD169

expression. Blimp-1[+] SPF cells were rounded and immotile and migrated slowly over small search areas in a subdiffusive pattern. In terms of search strategy, this behaviour has been suggested to increase the probability of finding nearby targets, particularly in areas where the target is abundant[31], as might occur for antigen trapped by SCS macrophages in the SPF. In contrast, a second population of Blimp-1[neg] cells were highly motile and migrated in a superdiffusive search pattern similar to that observed for naive and MBCs in the resting lymph node. Deconvolution of the secondary antibody response by single-cell RNA sequencing showed that this population expressed *Sell*, *Ccr6* and *Cd38* and shared gene expression signatures with MBCs. Thus, in addition to plasma cells, our studies reveal that the SPF is also a site for further MBC differentiation. It is interesting to speculate what 'targets' these MBCs are searching for in the SPF, whether it is antigen or T cell help or other signals required to complete their differentiation programme. These 'secondary' MBCs may be similar to 'early' MBCs generated outside the GC in primary antibody responses[56] and it remains to be seen how they contribute to B cell memory and humoral immunity.

A key question arising from our study is what determines the MBC fate decision to enter the SPF or GC. Recent studies have documented heterogeneity in the MBC pool in terms of immunoglobulin isotype and propensity to differentiate into GC B cells or plasma cells. In particular, it has been shown that IgM[+] MBCs preferentially re-enter GCs and IgG1[+] MBCs preferentially differentiate into plasma cells[57,58]. This imprinting of the MBC response also tracks with CD80 and PD-L2 expression[59], and may relate to increase expression of FOXP1 in IgM[+] MBCs, which has been shown to repress *PRDM1*, *IRF4* and *XBP1* to restrain plasma cell differentiation[60]. However, these are generalisations and indeed class-switched MBCs have also been shown to recycle through the GC and further mutate their BCR[61]. Nevertheless, it will be interesting to determine if immunoglobulin isotype also dictates the SPF versus GC cell fate in future studies. Our data from inducible deletion of MHCII on MBCs before antigen recall indicate that development of the SPF, similar to the GC, is dependent on T cell help. Interestingly, inhibition of BCR signalling with the small molecule Btk inhibitor ibrutinib selectively decreased the number of GC B cells without affecting the number of SPF cells. This suggests that, once established, the survival of B cells in the SPF is independent of Btk-mediated BCR signalling. Moreover, these data provide further support for the notion from the intravital imaging that SPF cells do not originate from the GC. This is confirmed by analysis of the single-cell gene expression trajectories by pseudotime and changes in the cellular composition of the response on day 5 and day 7.

In summary, we have identified the SPF as a novel immune response pathway for the reactivation of MBCs in immune animals. The efficient intrafollicular expansion and plasma cell differentiation of antigen-specific MBCs in SPF, where antigen and T cell help is concentrated, provides a previously unappreciated mechanism for the rapid kinetics of the secondary antibody response. Importantly, it is this rapid secretion of antibodies that protects the host from reinfection, well before LLPCs derived from secondary GCs are generated. These data establish the SPF as the seat of B cell memory, and as an evolutionarily conserved immune response pathway that may be targeted to improve human vaccine efficacy.

## Methods

**Mice.** SW_HEL mice expressing a knock-in BCR against hen egg lysozyme (HEL)[16] were maintained on a C57BL/6J or C57BL/6-SJL.Ptprc[a/a] congenic background. Mice expressing tdTomato were generated by crossing floxed tdTomato reporter (007914; B6.Cg-*Gt(ROSA)26Sor[tm14(CAG-tdTomato)Hze]*/J)[62] with *ROSA26*-Cre mice that ubiquitously expresses Cre recombinase (003474; B6.129S4-*Gt(ROSA) 26Sor[tm1Sor]*/J)[63]. Kaede (KD) transgenic mice (Tg(CAG-Kaede)#Kgwa)[17], Blimp-1[gfp] reporter mice[30] and tdTomato mice were crossed to SW_HEL mice to obtain fluorescent SW_HEL B cells on a C57BL/6J background. MHCII[fl/fl] mice (013181; B6.129 × 1-*H2-Ab1[tm1Koni]*/J)[64] were bred with *Cd79a*.CreERT2[65] and SW_HEL mice, and maintained on C57BL/6-SJL.Ptprc[a/a] congenic background. KD transgenic mice, CFP transgenic mice expressing cyan fluorescent protein under the β–actin promoter (Tg(ACTB-ECFP)1Nagy/J)[66] and Thy1.1 congenic mice (000406; B6.PL-*Thy1a*/CyJ)[67] were crossed to OT2 TCR transgenic mice (B6.Cg-Tg (TcraTcrb425Cbn/J))[18], and maintained on a C57BL/6 background. C57BL/6 and C57BL/6-SJL.Ptprc[a/a] congenic mice were purchased from Australian BioResources (Moss Vale, Australia). SAP-deficient mice (B6.129S6-Sh2d1atm1Pls/J)[68] were maintained on a C57BL/6J background. All mice were bred and maintained in specific pathogen-free conditions at Australian BioResources. Animal experiments were approved by the Garvan Institute of Medical Research/St Vincent's Hospital Animal Ethics Committee and complied with all relevant regulations.

**Adoptive cell transfer and immunisations.** OT2 T cells were enriched by negative depletion with biotinylated antibodies for anti-B220 clone RA3-6B2, anti-CD11b clone M1/70, anti-CD11c clone HL3, anti-CD8 clone 53-6.7, and SW_HEL B cells were enriched by negative depletion with biotinylated antibodies for anti-CD11b, anti-CD11c, anti-CD4 clone GK1.5, anti-CD43 clone S7 (all from BD Bisociences) and MACs anti-biotin magnetic beads (Miltenyi). Purity of CD4[+] V_α2[+] OT2 T cells was typically 70–80% and B220[+] HEL-binding SW_HEL B cells >99% as determined by FACs analysis. $2.5 \times 10^5$ CD4[+]Vα2[+] OT2 T cells and B220[+] HEL-binding SW_HEL B cells were adoptively transferred into age and sex matched 6–9-week-old C57BL/6 or SAP-deficient recipient mice. Recipient mice were immunised the next day by subcutaneous injection with 20 µg HEL-OVA in Sigma Adjuvant System (SAS, Sigma) in the lower flank and base of tail. For memory responses, mice that had been immunised were rested for at least 28 days and then re-challenged with 40 µg HEL-OVA in SAS injected subcutaneously in the lower flank and base of tail. HEL was conjugated to OVA_{323–339} peptide (CGGISQAV-HAAHAEINEAGR) (Mimotopes/Genscript) using the SMPH cross-linking agent Succinimidyl-6-([ß-maleimidopropionamido] hexanoate) (Thermo Fisher Scientific) to generate HEL-OVA[48].

**Ibrutinib treatment.** Ibrutinib (Chemietek) was dissolved in Captex 355 (ABITEC) at 1.56 mg/ml. Greater than 28 days after primary immunisation, mice were injected intraperitoneally twice each day with ibrutinib at a dose of 12.5 mg/kg, or vehicle control. Draining inguinal lymph nodes were harvested 0.5 day after last injection and analysed. In addition, mice were injected with ibrutinib or vehicle control as described above twice each day for 5 days, and re-immunised as described above on the third day of treatment. Draining inguinal lymph nodes were analysed 0.5 day after last injection, 3 days after re-immunisation. In addition, mice were re-immunised as described above and then, starting 2d after re-immunisation were administered ibrutinib or vehicle control as described twice each day for 2 days. Draining inguinal lymph nodes were then harvested on day 4 after re-immunisation, 0.5 day after last injection.

**Tamoxifen treatment.** Tamoxifen (Sigma-Aldrich) was dissolved in 100% ethanol at 20 mg/ml then diluted 1/10 in sunflower oil to a final concentration of 2 mg/ml. More than 28 days after primary immunisation, tamoxifen at 0.5 mg/mouse (250 µl) or vehicle control was injected intraperitoneally once daily for 5 days. Mice were rested for 1 week then draining inguinal lymph nodes were analysed or mice re-immunised as described above. If re-immunised, draining inguinal lymph nodes were then analysed at 4–5 days after re-immunisation.

**CD4+ T cell depletion.** More than 28 days after primary immunisation, 200 µg anti-CD4 (GK1.5, BioXcell) or isotype control (Rat IgG2b, BioXcell) was admi-

**Fig. 7** Molecular regulation of the subcapsular proliferative foci. **a** Enumeration of population 1, 2 and 3 in the secondary response following inducible deletion of MHCII. Data is representative of three experiments with $n = 6$ each group. Comparison between groups utilised one-tailed unpaired Student's *t*-test. **b** Enumeration of population 1, 2 and 3 in the secondary response following ibrutinib treatment after reactivation of MBCs. Data are representative of 2 experiments with $n = 5$ each group. Comparison between groups utilised one-tailed unpaired Student's *t*-test. Box plots show centre line as median, box limits as upper and lower quartiles, whiskers as minimum to maximum values. **c** Representative immunohistochemical analysis of draining cervical lymph node from a patient with head and neck cancer showing CD169 (brown) and CD138 (red) expression. Hematoxylin nuclear counterstain (blue). Scale bar = 100 µm

nistered intraveneously twice, two days apart. Mice were rested for 1 week then draining inguinal lymph nodes were analysed.

**Fiducial labelling for intravital two-photon microscopy.** CD169[+] SCS macrophages were marked in vivo by subcutaneous injection of fluorescently labelled anti-CD169 clone SER-4 (UCSF Hybridoma Core) 14–20 h before imaging. GCs were marked by by subcutaneous injection of fluorescently labelled anti-CD157 clone BP-3 (UCSF Hybridoma Core) 3–4 days before imaging[25]. Antibodies were labelled with either Alexa Fluor 555 or Alexa Fluor 680 protein labelling kit (Invitrogen), or CF680R protein labelling kit (Biotium) according to the manufacturers protocol.

**Intravital two-photon microscopy.** Intravital two-photon microscopy was performed as follows[69]. Briefly, anaesthesia was induced in the mouse with 100 mg/kg ketamine/5 mg/kg xylazine and maintained on anaesthesia on 1–2% isoflurane supplemented with 100% oxygen via a nose cone. The anesthetised mouse was kept warm on a customised heated SmartStage (Biotherm) set to 37 °C. The inguinal lymph node and intact inguinal ligament was mobilised in a skin flap and fixed on a base of thermal conductive T-putty (LairdTech) and PDMS polymer (Dow Corning). The cortical surface of the lymph node was exposed by microdisecction of the overlying fat and fascia layers under a stereomicroscope with low-level illumination. A small rubber O-ring was used to stabilise the meniscus and in some cases Immersol (Carl Zeiss) was also used. Imaging was performed on a Zeiss 7MP two-photon microscope (Carl Zeiss) powered by a Chameleon Vision II ultrafast Ti:Sa laser (Coherent Scientific). Images were acquired with a W Plan-Apochromat ×20/1.0 DIC (UV)Vis–IR water immersion objective. Excitation wavelengths used were 810, 870 and 920 nm. Fluorescent images were acquired with a LBF 760 and BSMP 760 to enable detection of far-red signals. Non-descanned detectors were SP 485 (blue; SHG and CFP), BP 500–550 (green; KD, GFP and CFP), BP 565–610 (red; PE and Alexa Fluor 555) and BP 640–710 (far-red; Alexa Fluor 680 or CF680R). Typically, 150 μm z-stacks were acquired in 3 μm steps. Image stacks were acquired at 30 s time intervals for 30 min or 60 cycles. Tiles of the whole lymph node were acquired with z-stacks of between 150 and 300 μm.

**Two-photon image analysis and mathematical modelling.** Raw image files were imported into Imaris (Bitplane) software package. The image intensities and thresholds were adjusted and a Guassian filter used to optimise signal to noise. Non-motile SCS macrophages were detected by the spot detection function and drift correction applied to correct for motion artefacts. Colocalisation between cells of interest was calculated by creating a COLOC channel and the COLOC intensity sum quantitated over the course of the movie. Cells were detected using the spot detection function and automatically generated tracks were manually verified. The proportion of cells in proximity (<20 μm) to SCS macrophages at each time point over the course of a movie was calculated using the Spots close to Surface XTension plugin (BitPlane). The distance of cells to the SCS macrophage surface was calculated using the Immune Atlas Imaris plugin (http://www.matebiro.com/software/immuneatlas). The speed of cells was extracted from the Imaris Statistics function. The arrest coefficient was calculated as the proportion of time a cell slowed down to less than 3 μm/min. The number and duration of interactions between T cells and B cells in the primary and secondary response was quantified and verified manually, and normalised for the length of the T cell track. In some experiments, surfaces of SCS macrophages (based on CD169 labelling), SPF (based on CD169 and B cells clustering around them in the subcapsular region) or GCs (based on B cells clustering deep in the follicle) were calculated and applied using the Imaris Surface function. The follicular mantle (FM) was defined as the region in between the GC and SPF, and individual tracks were then assigned in relation to the SPF, FM and GC boundaries. Cell track spatiotemporal position data were extracted from the Imaris Statistics function and motility parameters calculated and motility models fitted[27]. Briefly, data from replicate experiments were pooled together. Instantaneous velocities for time points within cell tracks of a given population were pooled together and then plotted as empirical cumulative distribution functions (the speed graphs). These show the proportion of the distribution lying at or below a given speed; a proportion of 0.5 represents the distribution's median. Net displacement was calculated as the spatial Euclidean distance between first and last observations of each track. A cell's meandering index is its net displacement as a proportion of total length travelled; a value of 1 indicates a straight line, zero indicates identical starting and ending positions. MSD for a given time interval was calculated, and the slope ascertained through linear regression. Time intervals up to 25% of the population's longest duration track were investigated, to avoid biasing towards slow, un-directional tracks that persist in the imaging volume. Displacement data for a given time interval were extracted from anywhere in the track timeseries, i.e. time intervals are not absolute from time zero or first observation. Track movements in the x and y axes were plotted for a representative sample of each population's cells. Track movements are shown relative to their starting positions, centred on the origin. Each plot shows up to 40 tracks, representing a uniform sampling of track displacements.

We make inferences of cell population motility dynamics by reproducing these processes in silico through agent-based simulation. Six putative models of cell motility were fitted to our in situ data, fully described in ref.[27]: Brownian motion, a

Lévy walk where agent speeds and durations are both drawn from Lévy distributions, and four correlated random walks (CRW). The CRWs differ in their treatment of all agents as statistically identical (homogeneous) or heterogeneous in their individual speed and turn dynamics (allowing agents to differ in their inherent speeds and directionalities), and either selecting translational and turn speeds independently of one another or imposing an inverse correlation between the two (inverse). Each motility model was realised in a 3D agent-based simulation representing cells as non-overlapping spheres. Simulated space is boundless, but spatiotemporal data are only recorded for agents residing within an artificial imaging volume of identical dimensions to that in situ. All random walk models, except Lévy walk, were simulated with a discretised time step of 30 s. Lévy walk employed a 3 s time step to better accommodate this model's variable walk durations. Simulated cell spatiotemporal locations were recorded every 30 s for all models, as with in situ imaging. Random walk model parameters best aligning model and in situ motility are obtained using multi-objective optimisation, i.e. Non-dominated Sorting Genetic Algorithm-II (NSGA-II)[70]. NSGA-II aligns simulated with in situ cell motility by minimising differences in (1) distributions of pooled cell instantaneous velocities (speeds), (2) distributions of pooled turn speeds, extracted similarly to instantaneous velocity, and (3) MSD slopes. No single set of model parameters perfectly aligns all three of these metrics, and NSGA-II maintains parameter value sets representing optimal tradeoffs. Each parameter value set is assigned a Λ value[27] quantifying its quality in terms of delivering equally good alignments across all metrics; low Λ values indicate superior reproduction of in situ motility. Standard model selection techniques such as the Akaike information criterion are inapplicable in this multi-objective, multi-metric context. Instead, we balance goodness of fit against model complexity by constraining the computational resource available for fitting; more complex models must offer tangible and readily realised improvement of capture over simple models within the computation available. NSGA-II parameters for model fitting employ a population size of 50 executed for at most 40 generations (fitting is terminated early if model overfitting is detected, see ref.[27]). Supplementary figures show model parameters Λ distributions and report Kolmogorov–Smirnov differences therein.

Time-lapse images were exported, compiled and annotated in Adobe AfterEffects (Adobe), and movies converted using HandBrake (https://handbrake.fr).

**FACS analysis and single-cell FACS sorting.** Draining inguinal lymph nodes were harvested, dissected free of fat and fascia, teased apart with microforceps and mashed through a 70 μm filter. Single-cell suspensions were then washed and Fc receptors blocked with unlabelled anti-CD16/32 clone 2.4G2 before staining. To detect HEL-binding B cells, cells were stained with saturating levels of HEL at 200 ng/ml, followed by HyHEL9 Alexa Fluor 647. For detection of HEL-binding IgG1[+] B cells, anti-IgG1 staining was performed first and followed by blocking with 5% mouse serum before subsequent staining for HEL-binding with HyHEL9, a mouse IgG1 monoclonal antibody. Antibodies used for staining are shown in Supplementary Table 1. For intracellular staining, cells were fixed with Fixation/Permeabilization buffer and antibodies stained in Permeabilization buffer (eBioscience). Antibodies used for intracellular staining were: Bcl6-PE (K112-91, BD Biosciences). Cells were filtered using 35 μm filter round-bottom FACS tubes (BD Biosciences) immediately before data acquisition on either an LSR II SORP or Fortessa (BD) and data analysed using FlowJo software (Tree Star, Inc.). Single-cell FACS sorting was performed on a FACS Aria III into a 384 well plate (Biorad) containing 3 μl of cell deposition buffer (0.05% 10× GeneAmp PCR buffer without MgCl (Life technologies); 0.9% Superase-In RNase Inhibitor (20 U/μl), (Life technologies); 0.9% RNasin® Plus RNase Inhibitor (40 U/μl), (Promega); Nuclease-free water (Qiagen)).

**Single-cell RNA sequencing.** Single-cell cDNA was generated and amplified using SMART-Seq® v4 Ultra® Low Input RNA Kit for Sequencing (SMARTer kit) (Scientifix) at half reaction volumes. For each sample, 0.5 μl of 1:2,500,000 Sequin RNA spike-in mixes[71] were incorporated in the master mix before RNA denaturation step and cDNA was amplified for 23 cycles. NexteraXT kit (Illumina) was used to generate libraries that were sequenced at 2×125 bp on the HiSeq2500 at the Kinghorn Center for Clinical Genomics (New South Wales, Australia).

**Bioinformatic analysis.** Raw sequence reads were quality control assessed using FastQC, and trimmed using Cutadapt. An index reference genome was created by concatenating the GRCm38 assembly of the *Mus musculus* genome and the RNA Sequins spike-in nucleotide sequence. Reads were pseudoaligned to this index and transcript abundance was quantified using kallisto[72]. Pre-processing, quality control and normalisation was performed using the scater package[73] in the R/Bioconductor statistical programming framework. Counts were converted to $\log_2$ (counts per million + 1). Genes with zero counts in all libraries were filtered from downstream analysis. Library size normalisation was performed using scran[74]. Three populations within the 122 single-cell dataset were identified using the unsupervised consensus-clustering package SC3[75]. Differential gene expression between the three clusters was determined using an integrated Bayesian hierarchical model, BASiCS[76] with a FDR cutoff of 0.1. A matrix of all the differentially expressed genes was then deconvoluted by NMF to identify

metagenes for each population[77]. Pseudotemporal ordering and single-cell trajectory analysis[35] was performed in Monocle2[78].

**Epifluorescence microscopy and quantification.** Lymph nodes were snap frozen in cryomolds with OCT (Tissue Tek). Overall, 7 μm sections were cut using a CM3050S cryostat (Leica), transferred to PolySine glass slides and air-dried. Cut sections were fixed in ice-cold acetone, dried and blocked with 30% horse serum (Invitrogen), 3% BSA in PBS. Antibodies used are listed in Supplementary Table 1. Sections were incubated with 200 ng/ml of HEL and detected with polyclonal rabbit anti-HEL IgG (Rockland Immunochemical Inc) followed by goat anti-rabbit FITC (Southern Biotech). Slides were visualised on a Leica DM5500 microscope. Images were compiled and brightness and contrast adjusted in Adobe Photoshop. For quantification, raw single channel image files were imported into Imaris software and manual surfaces created corresponding to the SPF region (directly underneath the capsule in the B220 + B cell follicle), the GC region (containing the CD35 + LZ and Ki67 + DZ), and the FM in between the SPF and GC. These surfaces were masked for channels of interest and spots on either total B cells (B220+) or HEL-binding cells (HEL+) were then detected using the spot detection function and staining intensity for HEL, Ki67 or Bcl6 calculated.

**Human immunohistochemistry.** Archived paraffin-embedded human cervical lymph nodes from patients with head and neck cancer who had undergone lymph node dissection as part of their routine patient care were stained for anti-human CD169 (clone HSn 7D2, In Vitro Technologies) and CD138 (clone B-A38, Cell Marque) according to the manufacturers protocols.

**Quantification and statistical analysis.** Data were analysed with Prism software (GraphPad). For comparison between two normally distributed groups a one-tailed unpaired Student's $t$-test with Welch's correction was used, and for more than two groups we used one-way ANOVA with Tukey's correction for multiple comparisons. Non-parametric data were analysed by Mann–Whitney $U$ test. Differences between multiple paired measurements were analysed by the Wilcoxon signed-rank test. $*p < 0.05$, $**p < 0.01$, $***p < 0.001$ and $****p < 0.0001$. The scientific ($D$) and statistical ($p$ value) significance of differences in motility parameter distributions (speed, displacement and meandering indexes) was assessed through the Kolmogorov–Smirnov statistic. This non-parametric test quantifies the maximum distance between two empirical cumulative distribution functions, and is thus sensitive to differences in distributions' locations and shapes. $D$ is the maximum difference in the proportions of the two distributions lying at or below a given value, and ranges from 0 (distributions are identical) to 1 (distributions do not overlap).

**Data availability.** The single-cell RNA sequencing data are publicly available from the GEO, accession number GSE 109984. Relevant data are available from TGP upon request.

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

## Acknowledgements

We thank Stephen Nutt, Jason Cyster and Julie Zikherman for helpful suggestions and discussion. We thank Tony Basten and Jonathan Sprent for critical reading of the manuscript. I.M. is supported by Australian Postgraduate Award. S.G.T., R.B. and A.D.K. are supported by Fellowships from the NHMRC. This work was funded by the NHMRC Project Grants 1124681 (T.G.P., C.S.M., F.L., J.Z.) and 1139865 (T.G.P., C.S.M.). M.N.R. is supported by the Judith and David Coffey Life Lab.

## Author contributions

I.M. and T.G.P. conceived the study and designed the research. A.N., I.M. and T.G.P. designed the single-cell experiments. I.M., A.N. and W.H.K. performed single-cell RNA sequencing. A.N. analysed the single-cell RNA sequencing. D.B., K.B., C.Y. and J.H. provided technical support. C.S.M., C.M.L.M., F.L., J.Z., A.D.K., S.G.T., P.I.C. and R.B. provided critical advice on experimental design, analysis and interpretation. M.B. provided image analysis tools and advice. I.M. and M.N.R. performed image analysis. M.N.R. performed mathematical modelling. G.G. and A.P. performed human immunohistochemistry. I.M., A.N., M.N.R. and T.G.P. wrote the manuscript. All authors have read and approved the manuscript.

## Additional information

**Competing interests:** The authors declare no competing interests.

