## [Peer Review File · Nature Communications]

Reviewers' comments:

Reviewer #1 (Remarks to the Author):

This manuscript describes the B cell response to recall immunization in the LN at the microanatomical level. The authors find that ~1 month after immunization, B cells reside in a region immediately beneath the subcapsular sinus in close proximity to macrophages, and that upon recall immunization these cells proliferate in this location as well as in GCs, and follow the three fates that are expected to derive from memory B cell reactivation (GC, plasmablast, and memory cell regeneration). Although this is an interesting description of a phase of the immune response that is not completely understood, my feeling is that it does not contribute much information that was not previously known. Memory cells have been shown to reside in a different area compared to naïve cells; proliferation in the outer follicle at days 4-5 was previously described by Manser and colleagues (see below); and the differentiation of MBCs into distinct fates upon recall has also been described in several papers in the last decades. Enthusiasm for this manuscript was further decreased by certain methodological issues (see below), which could be addressed by further experiments:

1. Figure 1: How sure are the authors that the cells they study are true MBCs? Only Figure 1A is done at 70 days, the rest is 28-30, when the primary response is not sure to have been resolved (GCs often last longer than 1 month even with protein in adjuvant). BrdU incorporation or Ki67 staining could be used to ascertain that these are indeed resting B cells; FACS or histology could be used to show that GCs are gone at this point.
2. Figure 1C: the authors show that >50% of B cells are within 20um of the follicle, which seems more compact than one would expect given the size of a normal B cell follicle. Does the two-photon system used allow for imaging at greater depths? (the interior of the LN in movie S1 looks conspicuously empty).
3. Figure 1D: The interaction of MBC with macrophages requires further validation. Isn't some degree of colocalization inevitable between cells in the same area? Does colocalization mean that these cells are interacting somehow via surface receptors? Comparing the interactions with those made by naïve B cells could help establish whether MBC-macrophage interactions are indeed significant. Determining whether MBC slow down when in contact with macrophages could help as well.
4. Figure 1E: The definition of the non-draining control LN is not clear. Is this the ipsilateral or the contralateral popliteal LN? The ipsilateral pLN would be a poor control since it was not ruled out that there is some draining of antigen to the ipsilateral pLN from the priming. Looking for GCs in the pLN would help settle this.
5. Figure 1F: Authors should show FACS plots for the photoactivation experiment to show how good the separation between photoactivated and non-photoactivated cells is.
6. Figure 3A-B: The outer follicle area has previously been described as the site of B cell proliferation at day4-5 of the immune response (Coffey, Alabayev & Manser, *Immunity*, 2009, which the authors should cite). Therefore the concept of the SPF as a new anatomical compartment is not entirely novel. Co-transfer of naïve S_Whel cells prior to boost could settle whether MBC are occupying a separate niche from that previously described by Manser.
7. Figure 3C: The more novel claim seems to be that T cell-B cell interactions take place at the SPF site, but this description is not strongly supported: only one anecdotal time series and graph are shown, and multiple tracks/movies are not quantified in any way. The usual controls for interaction, such as naïve polyclonal B cells, are also missing. Furthermore, it is not clear that any of the data in figure 3 come from more than one movie.

8. Figure 4: clear GCs with two zones are already formed at 3 days post boost, which seems incredibly fast. Are the authors sure these GCs were not there already prior to boost (see comment no. 1 above).
9. Figure 6: It is not clear what the contribution of the single-cell RNA-seq is when all three populations are previously known and can be identified by known FACS markers. There is also no analysis of how the proportions of each cell type identified by flow match those identified by RNA-seq.
10. Figures 6 and 7A: another interpretation of the 2 non-GC populations is that the MBC-like cells are on their way to becoming plasmablasts/plasma cells. Is this population gone if a later time point is analyzed?
11. Figure 7B: does tamoxifen treatment reduce the number of MBC in the LN prior to boosting? In other words, is T cell help required to maintain MBC numbers? If these are truly resting MBC, their maintenance should not be dependent on ongoing T cell help.
12. Figure 7C: Being resistant to ibrutinib treatment is not equal to being BCR-independent, since there are multiple BCR-related pathways that are not downstream of Btk. I would rephrase this claim to reflect this limitation.

Reviewer #2 (Remarks to the Author):

The study by Moran et al. convincingly demonstrated the possibility that subcapsular regions in lymph nodes are niches for survival, although this study has not addressed directly whether this region acts indeed as a niche. Upon re-activation with a specific antigen, MBCs proliferate and differentiate into plasma cells in the regions with help from memory T cells. The authors went on to analyze the fate of re-activated MBCs and dissected three populations, plasma cells, secondary GC B cells, and MBCs, which have distinct requirement of BCR signaling for their survival. Overall, the data are of high quality.

But, I have three concerns about this manuscript.

First, although authors carefully analyzed MBC before secondary immunization, they should describe the status of plasma cells and GC cells before secondary immunization. Particularly, since GC B cells sometimes survive unexpectedly for long time, they should carefully analyze the GC status. Otherwise, it is very hard to exclude the possibility that secondary GCs observed in fig. 6 and 7 might be at least to some extents are derived from reactivation of remaining primary GCs.

Second, between primary and secondary responses, what is difference of T-B interaction durations? I think that this could be addressed by authors, being important information.

Finally, the concern is the conclusion regarding the role of BCR signaling for survival of SPF B cells (MBCs) (Fig 7C). Probably, such apparently secondary memory B cells are derived from activated primary memory B cells. So, if inhibited, initial activation of primary memory B cells might be inhibited or subsequent secondary memory B cell survival might be inhibited. The authors' experiments did not distinguish these two possibilities. I suggest authors to improve experimental designs for this point, although critically distinguishing these possibilities is difficult. Furthermore, the critical role of BCR signaling for MBC survival has been previously suggested by a study with inducible knock-out of Syk tyrosine kinase from MBCs (Ackermann et al., J. Immunol. 2015, 194: 4650-4656). The authors should discuss the discrepant conclusion. In the same line, because survival of resting mature (naïve) B cells depends on BCR signaling (Lam et al., 1997,

Kraus et al., 2004), I suggest to include naïve SWHEL B cells as a control to show that the current dose of Ibrutinib is enough to block survival of naïve B cells, but not of MBCs.

Point-by-point response to reviewers

Reviewer 1

1. Figure 1: How sure are the authors that the cells they study are true MBCs? Only Figure 1A is done at 70 days, the rest is 28-30, when the primary response is not sure to have been resolved (GCs often last longer than 1 month even with protein in adjuvant). BrdU incorporation or Ki67 staining could be used to ascertain that these are indeed resting B cells; FACS or histology could be used to show that GCs are gone at this point.

Response: We agree with the reviewers that in some systems GCs can persist for longer than 1 month. This is a critical point and we have therefore performed several experiments and added multiple lines of supporting data to show these cells are bona fide MBCs. Specifically:

1. the cells are Ki67 negative and no longer proliferating by histology (**Supplementary Figure 1A**);
2. the cells by FACS do not express GC markers but instead express classic MBC markers. For example, they are CD38 high (not CD38 low) (**Supplementary Figure 1B**) and express CCR6 (**Supplementary Figure 2**);
3. the cells are independent of T cell help as shown by T cell depletion with anti-CD4 monoclonal antibody (**Supplementary Figure 1C**) and inducible depletion of MHCII (**Supplementary Figure 1D**).
4. the cells are independent of BCR signaling as shown by their persistence following ibrutinib treatment (please also see response to Reviewer 2 below re: ibrutinib) (**Supplementary Figure 1E**).

Changes to manuscript:

Data. **Supplementary Figure 1.** A. Histology of Ki67 and HEL-binding. B. FACS of CD38 versus Fas. C. T cell depletion with anti-CD4 treatment. D. Inducible depletion of MHCII with tamoxifen treatment on MBCs. E. Ibrutinib treatment.

Text. Main text added: Mice were analysed >28 days later when the primary antibody response has resolved and antigen-specific cells are no longer proliferating (Supplementary Figure 1). After this timepoint there are no persisting GCs, as demonstrated by fluorescence-activated cell sorting (FACS) analysis (Supplementary Figure 1). MBCs are able to survive independent of antigen-derived BCR signals²⁵ and T cell help^{26, 27}, unlike GC B cells which are dependent on both^{28, 29}. T cell depletion experiments and inducible deletion of MHCII in responding B cells had no impact on the survival of these cells >28 days after primary immunisation (Supplementary Figure 1). Furthermore, inhibition of BCR signaling with the small molecule ibrutinib also did not impact on

their survival (Supplementary Figure 1). Finally, these cells consisted of IgM⁺ and IgG⁺ cells that expressed Fas, CD80, CD86, PD-L2, CCR6, CD69, CD62L, EB12 and CXCR3, but not S1PR1 (Supplementary Figure 2), consistent with the phenotype of murine MBCs³⁰. Taken together, these data confirm that antigen-specific cells persisting >28 days after primary immunisation were bona fide MBCs and not GC B cells.

2. Figure 1C: the authors show that >50% of B cells are within 20µm of the follicle, which seems more compact than one would expect given the size of a normal B cell follicle. Does the two-photon system used allow for imaging at greater depths? (the interior of the LN in movie S1 looks conspicuously empty).

Response: We apologise if this gave the impression that the follicle is compact, but Fig. 1C is related to dynamic intravital imaging of the position of individual B cells over the timecourse of the movie when we imaged the region shown in Fig. 1B. In this analysis, we use the Spots Close to Surface Xtension in Imaris to track the position of the same cells over time as they migrate in the follicle. Thus, slow moving cells that are close to the SCS macrophage surface are captured multiple times as spots in this compressed volume, whereas fast moving cells deeper in the follicle are captured as fewer spots spread out over larger volume. To provide a more intuitive measure, we have also quantified the positions of naïve and MBCs from the capsule surface. This shows that, on average, MBCs are positioned much closer to the SCS macrophages than naïve B cells. Furthermore, this analysis shows that our imaging system is able to easily detect cells at depths up to 300µm below the capsule.

Changes to manuscript:

Data. Supplemental Figure 3. A. Plots of actual distance of naïve B cells and MBCs from the capsule.

Text. Main text: Analysis of whole-mount lymph nodes by two-photon microscopy (TPM) 70 days later revealed that while B cells were detectable in the follicle as deep as 300µm below the surface of the capsule (Supplementary Figure 3), antigen-specific SW_{HEL} donor-derived B cells were preferentially positioned in the outer B cell follicle, where they were closely associated with CD169⁺ subcapsular sinus (SCS) macrophages (Figure 1A and Supplementary Movie 1).

3. Figure 1D: The interaction of MBC with macrophages requires further validation. Isn't some degree of colocalization inevitable between cells in the same area? Does colocalization mean that these cells are interacting somehow via surface receptors? Comparing the interactions with those made by naïve B cells could help establish whether MBC-macrophage interactions are indeed

significant. Determining whether MBC slow down when in contact with macrophages could help as well.

Response: Colocalisation is not inevitable between cells in the same area as they occupy different pixels. The colocalization analysis in Imaris detects overlapping pixels from both cell types and arises due to cell surface contacts and not chance, implying surface receptors such as adhesion molecules are involved. We and others have previously shown similar scanning behaviour by naïve B cells as they surveilled SCS macrophages for antigen (Junt et al Nature 2007, Carrasco et al Immunity 2007, Phan et al Nat Immunol 2007). As requested, we have performed additional analyses to show that MBCs have a greater arrest coefficient compared to naïve B cells. That is, MBCs spend a greater proportion of a time at a speed of less than 3µm/min, which would be consistent with them slowing down to scan SCS macrophages. In addition, we have analysed the speed of MBCs as they approached the SCS macrophages. As predicted by Reviewer 1 this showed that MBCs were significantly slower when in proximity to SCS macrophages than when far away, again suggesting MBCs are interacting with these macrophages and slowing down in the process.

Changes to manuscript:

Data. Figure 1 E. Speed of MBCs based on distance of MBCs from SCS macrophages.

Data. Figure 1 F. Arrest coefficient of naïve and MBCs.

Text. Main text added: In addition, MBCs in close proximity (<20µm) to SCS macrophages had a decreased average speed compared to MBCs further away (>20µm) from SCS macrophages (Figure 1E) and MBCs had an increased arrest coefficient compared to naïve B cells (Figure 1F). These findings further support the conclusion that MBCs in the subcapsular region are scanning SCS macrophages in search of antigen.

4. Figure 1E: The definition of the non-draining control LN is not clear. Is this the ipsilateral or the contralateral popliteal LN? The ipsilateral pLN would be a poor control since it was not ruled out that there is some draining of antigen to the ipsilateral pLN from the priming. Looking for GCs in the pLN would help settle this.

Response: As mice were immunised on both sides, the popliteal LNs analysed were ipsilateral to the side of immunisation. We have previously performed injections of India ink to track the lymphatic flow from subcutaneous immunisations and these experiments showed that injections to the base of the tail and flank do not flow retrograde to the ipsilateral popliteal lymph node. Nevertheless, we have looked for GCs in the ipsilateral popliteal LN as requested and this showed that there were no GCs evident from the priming. This is presented as **Additional Reviewer Data 1**

which showed that all donor-derived cells are CD38^{hi} and therefore unlikely to be GC B cells which have a CD38^{lo} phenotype.

Changes to manuscript:

Data. Additional Reviewer Data 1A. FACS of CD38 expression on donor-derived cells in the ipsilateral popliteal LN.

5. Figure 1F: Authors should show FACS plots for the photoactivation experiment to show how good the separation between photoactivated and non-photoactivated cells is.

Response: Thank you for this suggestion. We have now added a FACS plot to show the photoconverted Kaede red can be clearly separated from the non photoconverted Kaede green cells.

Changes to manuscript:

Data. Figure 1H. FACS plots showing photoconverted Kaede red compared to non-photoconverted Kaede green

6. Figure 3A-B: The outer follicle area has previously been described as the site of B cell proliferation at day 4-5 of the immune response (Coffey, Alabayev & Manser, Immunity, 2009, which the authors should cite). Therefore the concept of the SPF as a new anatomical compartment is not entirely novel. Co-transfer of naïve SWhel cells prior to boost could settle whether MBC are occupying a separate niche from that previously described by Manser.

Response: There are key differences between what Coffey et al. describes and what we show and thus we respectfully disagree that the SPF anatomical compartment has been described before. Firstly, Coffey et al. describes the systemic immune response in the spleen rather than the draining immune response in the lymph node. Secondly, and perhaps most importantly, Coffey et al. describe pre-GC B cells that migrate to the T-B border and then to the outer follicle area before entering GCs in the primary response. In contrast, what we are describing is MBCs pre-positioned in the outer follicle subcapsular region that do not need to migrate to the T-B border to receive T cell help as memory T cells are also pre-positioned in this region. Moreover, re-activated MBCs in the subcapsular region can locally proliferate and differentiate into plasma cells rather than migrating to the GC to differentiate into post-GC plasma cells. Thus, we are describing a site of both activation, T cell help and differentiation rather than a site solely of proliferation. While we do not disagree with the findings of Coffey et al., we think their work describes a different phase of the immune response.

Changes to manuscript:

Text. Added new text in **Discussion:** While proliferation of B cells at the perimeter of the follicles in the spleen had been previously described in the primary antibody response³⁸, under these circumstances naïve B cells are still required to migrate to the T-B border to acquire T cell help and be licensed to differentiate into plasma cells.

7. Figure 3C: The more novel claim seems to be that T cell-B cell interactions take place at the SPF site, but this description is not strongly supported: only one anecdotal time series and graph are shown, and multiple tracks/movies are not quantified in any way. The usual controls for interaction, such as naïve polyclonal B cells, are also missing. Furthermore, it is not clear that any of the data in figure 3 come from more than one movie.

Response: In the manuscript we presented a representative time series from one of multiple examples from more than 4 movies from independent experiments. However, we appreciate the Reviewer's comment and have extended the analyses as described below with additional experiments.

As shown by Liu et al. Nature 2015 and Shulman et al. Science 2014, T-B interactions in the GC are brief and transient, unlike the prolonged stable conjugates formed at the T-B border. Consistent with this we observe activated T and B cells in the SPF also make brief transient contacts. It has previously been shown that naïve polyclonal B cells interact with Tfh cells also contact and transiently interact with Tfh cells in the follicle (Xu et al. Nature 2013); therefore we feel that this may not be the most appropriate control. Therefore, we have elected to image the interactions on day 5 of the primary response, when activated T cells and activated B cells have colonised the follicle and migrated to the subcapsular region. We have now extended the analysis to provide additional quantification of T-B interactions from 4 independent movies in the secondary response, in comparison to 2 independent movies from the same time point in the primary response. This allowed measurement of both the average number of B cells that a T cell interacts with in a 5 minute time period, as well as the average duration of each contact. Interestingly, in both the primary and secondary response, activated T and B cells interact similarly to activated B cells and Tfh cells in the GC. These data further support the notion that the subcapsular region is a previously unappreciated site of T cell-B cell interactions.

Changes to manuscript:

Data. Figure 3D: number of times a T cells interacts with a B cell in 5 minutes in the primary and secondary response. **Figure 3E:** duration of T-B interactions in the primary and secondary response.

Text. Main text p.6 added: To further characterise these T-B interactions, we also imaged the primary antibody response on day 5, when both T and B cells have colonised the follicle³¹. These data showed that the T-B interactions were similar in number (Figure 3D) and duration (Figure 3E) between the primary and secondary responses.

8. Figure 4: clear GCs with two zones are already formed at 3 days post boost, which seems incredibly fast. Are the authors sure these GCs were not there already prior to boost (see comment no. 1 above).

Response: As outlined in the response to Comment 1 we have now definitively shown using multiple experimental approaches that there are no GCs present prior to boost. This rapid generation of secondary GCs is well characterised in several experimental systems and is a hallmark of immunological memory.

9. Figure 6: It is not clear what the contribution of the single-cell RNA-seq is when all three populations are previously known and can be identified by known FACS markers. There is also no analysis of how the proportions of each cell type identified by flow match those identified by RNA-seq.

Response: We apologise for not making clear the contribution of the scRNAseq. While in retrospect it is possible to say that all three populations are known and could be identified by FACS, this was not possible until the scRNAseq was performed and the three populations deconvoluted by NMF, the unique metagene contributions identified and the suspected cell populations confirmed by GSEA. Furthermore, scRNAseq identified key genes which were then used to identify the cells in the SPF by FACS, namely Sca-1 and CXCR3. Neither of these are canonical markers which could have been predicted without the scRNAseq. Furthermore, scRNAseq allows us to perform additional analyses such as diffusion pseudotime (see response to Comment 10 below) to determine the trajectories of the transcriptional states and infer lineage relationships.

We have also added the proportion cells in each population identified by scRNAseq in **Additional Reviewer Data 1B**. This shows that the proportions are similar, with some enrichment for plasma cells compared to MBC-like SPF B cells possibly due to the higher level of transcriptional activity in plasma cells that are actively secreting antibodies compared to quiescent MBCs.

Changes to manuscript:

Data. Additional Reviewer Data 1B. Proportion of cells in population 1, 2 and 3 from scRNAseq.

10. Figures 6 and 7A: another interpretation of the 2 non-GC populations is that the MBC-like cells are on their way to becoming plasmablasts/plasma cells. Is this population gone if a later time point is analyzed?

Response: As requested, we performed additional FACs analysis at a later time point of the response, day 7, and found the proportion of MBC-like SPF cells was similar between day 5 and day 7. In contrast, the proportion of the plasma cell population significantly decreased between day 5 to day 7. This shows the MBC-like cells do not disappear at a later time point and, as the plasma cell population has already started to decline at day 7, are not likely to be on their way to becoming plasma cells. To further explore the relationship between the three populations we performed pseudotime analysis using Monocle2. This showed that population 2 (plasma cells) and 3 (SPF B cells) were closely related and spaced together in pseudotime, compared to population 1 (GC) which was much further apart. The branching suggests that population 3, not surprisingly, can give rise to population 2.

Changes to manuscript:

Data. Figure 6H: Pseudotime analysis of the scRNAseq data.

Data. Figure 6 I: Proportion of population 1, 2 and 3 by FACs analysis at day 5 and day 7.

Text. Main text p.8 added:

Pseudotime analysis of lineage trajectories

To further characterise the relationship between these three cell populations, we performed pseudotime analysis to infer their lineage trajectories⁴¹. This showed that a large distance separated population 1 from population 2 and 3, which were closely spaced in pseudotime (Figure 6H). Thus, in accordance with the imaging data, responding B cells can be separated into GC and SPF cells, and the latter separated again into plasma cell and SPF cells with MBC-like properties. As plasma cells are terminally differentiated, this pseudo-temporal ordering also suggests the population 2 plasma cells can arise from a subset of population 1 SPF B cells. To further determine the origin of population 2 plasma cells, we compared the secondary antibody response on day 5 and day 7. This showed that, consistent with their expression of CXCR3, the plasma cells were short-lived and decreased between day 5 and day 7 (Figure 6I). At the same time there was an increase in the proportion of population 1 GC B cells and no change in the proportion of population 3 SPF B cells.

This suggests that the plasma cells in the SPF are unlikely to be derived from the secondary GC, as one would expect a concomitant increase in plasma cells as the GC expands and matures⁴². Thus, the imaging and single cell RNA sequencing confirm that short-lived plasma cells are generated from MBCs in the SPF.

11. Figure 7B: does tamoxifen treatment reduce the number of MBC in the LN prior to boosting? In other words, is T cell help required to maintain MBC numbers? If these are truly resting MBC, their maintenance should not be dependent on ongoing T cell help.

Response: We thank the reviewer for this important question. Please see response to Comment 1 above. We have performed tamoxifen treatment prior to boosting and shown that this does not impact on MBC numbers (new **Supplementary Figure 1**). In addition, we have also depleted CD4+ T cells and shown that this also does not impact on MBC numbers. Thus, we can conclude that these are truly resting B cells as they are not dependent on T cell help. (We have also shown they are Ki67 negative and have the phenotype of MBC and not GC B cells by FACS and resist ibrutinib treatment. See Comment 1 above.)

Changes to manuscript:

Data. Supplementary Figure 1C: Effect of anti-CD4 antibody treatment on number of MBCs.

Data. Supplementary Figure 1D: Effect of tamoxifen treatment on inducible MHCII^{fl} MBCs.

Text. Main text p.4 added Mice were analysed >28 days later when the primary antibody response has resolved and antigen-specific cells are no longer proliferating (Supplementary Figure 1). After this timepoint there are no persisting GCs, as demonstrated by fluorescence-activated cell sorting (FACS) analysis (Supplementary Figure 1). MBCs are able to survive independent of antigen-derived BCR signals²⁵ and T cell help^{26, 27}, unlike GC B cells which are dependent on both^{28, 29}. T cell depletion experiments and inducible deletion of MHCII in responding B cells had no impact on the survival of these cells >28 days after primary immunisation (Supplementary Figure 1). Furthermore, inhibition of BCR signaling with the small molecule ibrutinib also did not impact on their survival (Supplementary Figure 1). Finally, these cells consisted of IgM⁺ and IgG⁺ cells that expressed Fas, CD80, CD86, PD-L2, CCR6, CD69, CD62L, EB12 and CXCR3, but not S1PR1 (Supplementary Figure 2), consistent with the phenotype of murine MBCs³⁰. Taken together, these data confirm that antigen-specific cells persisting >28 days after primary immunisation were bona fide MBCs and not GC B cells.

12. Figure 7C: Being resistant to ibrutinib treatment is not equal to being BCR-independent, since there are multiple BCR-related pathways that are not downstream of Btk. I would rephrase this claim to reflect this limitation.

Response: We have rephrased this to say Btk-mediated BCR signalling.

Changes to manuscript:

Text. Changed text throughout to say Btk-mediated BCR signaling eg **SPF B cell and plasma cell survival is independent of Btk-mediated BCR signaling**

Reviewer 2

First, although authors carefully analyzed MBC before secondary immunization, they should describe the status of plasma cells and GC cells before secondary immunization. Particularly, since GC B cells sometimes survive unexpectedly for long time, they should carefully analyze the GC status. Otherwise, it is very hard to exclude the possibility that secondary GCs observed in fig. 6 and 7 might be at least to some extents are derived from reactivation of remaining primary GCs.

Response: As outlined in the response to Comment 1 we now provide compelling evidence that GCs are not present prior to boost. Specifically:

1. the cells are Ki67 negative and no longer proliferating by histology (**Supplementary Figure 1A**);
2. the cells by FACS do not express GC markers but instead express classic MBC markers. For example, they are CD38 high (not CD38 low) (**Supplementary Figure 1B**) and express CCR6 (**Supplementary Figure 2**);
3. the cells are independent of T cell help as shown by T cell depletion with anti-CD4 monoclonal antibody (**Supplementary Figure 1C**) and inducible depletion of MHCII (**Supplementary Figure 1D**).
4. the cells are independent of BCR signaling as shown by their persistence following ibrutinib treatment (please also see response to Reviewer 2 below re: ibrutinib) (**Supplementary Figure 1E**);

Second, between primary and secondary responses, what is difference of T-B interaction durations? I think that this could be addressed by authors, being important information.

Response: We performed additional experiments to quantify T-B interactions from four independent movies in the secondary response in comparison to two independent movies from the

same time point in the primary response and found a similar average T-B interaction duration between the primary and secondary response. These data show that in the subcapsular region, T-B interactions were similar in primary and secondary antibody responses. See also response to Reviewer 1 Comment 7 above.

Changes to manuscript:

Data. Figure 3D: number of times a T cells interacts with a B cell in 5 minutes in the primary and secondary response. **Figure 3E:** duration of T-B interactions in the primary and secondary response.

Text. Main text p.6 added: To further characterise these T-B interactions, we also imaged the primary antibody response on day 5, when both T and B cells have colonised the follicle³¹. These data showed that the T-B interactions were similar in number (Figure 3D) and duration (Figure 3E) between the primary and secondary responses.

Finally, the concern is the conclusion regarding the role of BCR signaling for survival of SPF B cells (MBCs) (Fig 7C). Probably, such apparently secondary memory B cells are derived from activated primary memory B cells. So, if inhibited, initial activation of primary memory B cells might be inhibited or subsequent secondary memory B cell survival might be inhibited. The authors' experiments did not distinguish these two possibilities. I suggest authors to improve experimental designs for this point, although critically distinguishing these possibilities is difficult, Furthermore, the critical role of BCR signaling for MBC survival has been previously suggested by a study with inducible knock-out of Syk tyrosine kinase from MBCs (Ackermann et al., J. Immunol. 2015, 194: 4650-4656). The authors should discuss the discrepant conclusion. In the same line, because survival of resting mature (naïve) B cells depends on BCR signaling (Lam et al., 1997, Kraus et al., 2004), I suggest to include naïve SWHEL B cells as a control to show that the current dose of Ibrutinib is enough to block survival of naïve B cells, but not of MBCs.

Response: We appreciate the reviewers insight and agree that this is complicated and distinguishing these possibilities is difficult. In order to test this, we initially looked at the effect of ibrutinib treatment on survival of resting MBCs and found there was no statistically significant difference in the number of MBCs, suggesting that Btk-mediated BCR signalling was not required for resting MBC survival (**Supplementary Figure 1**). This is consistent with classic experiments from the Rajewsky lab (Maruyama et al. Nature 2000) showing that MBC survival is independent of antigen-derived BCR signals. However, as pointed out by Reviewer 2, it contrasts with Ackermann et al. J Immunol 2015 which showed that inducible deletion of Syk impaired MBC survival. This contradiction may be explained by the fact that Btk mediates active antigen-derived BCR signals,

whereas Syk mediates both active and ‘tonic’ BCR signals independent of antigen. These other signals could be generated by a number of receptors in addition to the BCR, including Fc receptors, integrins and C-type lectins (Mocsai et al Nat Rev Immunol 2010). Consistent with this, ibrutinib has been shown to inhibit the survival of pre-GC B cells, which have active BCR signaling, but not naïve or MBCs in patients with chronic lymphocytic leukemia (Sun et al Blood 2015). It is also consistent with a recent paper Nyhoff et al. J. Immunol 2018 which showed that Btk is dispensable for mature B cell survival. Our data also shows that ibrutinib treatment does not affect the survival of mature naïve B cells (**Additional Reviewer Data 1C**).

We have also performed additional experiments (**Supplementary Figure 8**) to distinguish between the effect of ibrutinib in blocking the initial activation of MBCs and the survival of reactivated MBCs. Not surprisingly this showed a decline in all 3 populations, including population 3 which was not evident when ibrutinib was administered after MBCs were reactivated, indicating that while Btk was required for reactivation of MBCs, it was not required for the survival in the SPF once they had been reactivated.

Changes to manuscript:

Data. Supplementary Figure 1: Effect of ibrutinib on MBC survival.

Data. Supplementary Figure 8: Effect of ibrutinib on reactivation of MBCs.

Text. In Discussion p.10 added: To study the dynamics of MBC localisation and reactivation in situ, we have established an experimental system in which, after 28 days, the primary antibody response has resolved and there are no persisting GCs. This was confirmed by the absence of Ki67 staining, the CD38^{hi} phenotype of the MBCs and their survival independent of Btk-derived BCR signals and T cell help. The survival of MBCs in our system following ibrutinib treatment is in agreement with their antigen-independence²⁵, but contrasts with recent data showing MBC survival is dependent on Syk. However, Btk only mediates active antigen-derived BCR signals, whereas Syk mediates both active and ‘tonic’ BCR signals independent of antigen, possibly downstream of its activation by Fc receptors, integrins or C-type lectins⁵¹, and this may account for the discrepancy. Consistent with this, Btk has recently been shown to be dispensable for mature B cell survival⁵² and ibrutinib treatment has been shown to deplete pre-GC B cells but not MBCs in patients with chronic lymphocytic leukemia⁴³.

Text. Main text p.9 Since ibrutinib treatment had no effect on survival of MBCs (Supplementary Figure 1), this suggests that Btk-mediated BCR signaling is required for initial activation of MBCs but not for the survival of MBCs in the SPF, whereas it is required for survival of GC B cells. This is consistent with clinical data showing that ibrutinib depletes pre-germinal centre B cells but not isotype-switched memory B cells in patients with chronic lymphocytic leukemia⁴³. Since ibrutinib

may affect the initial activation of MBCs, we also administered ibrutinib before and during activation of the secondary response (Supplementary Figure 8). As expected, this resulted in a significant reduction in the number of GC B cells as well as SPF B cells and a statistically non-significant reduction in the number of plasma cells. Thus, Btk-mediated BCR signaling is required for MBC reactivation and the survival of GC B cells, but not the survival of SPF B cells and plasma cells.

REVIEWERS' COMMENTS:

Reviewer #1 (Remarks to the Author):

The authors have addressed most of my concerns with additional data and/or clarifications.

Reviewer #2 (Remarks to the Author):

Reasonably responded to my concerns, I can recommend its publication in N.C.